# Phonemic segmentation of narrative speech in human cerebral cortex

Xue L. Gong [1] ✉, Alexander G. Huth[2], Fatma Deniz[3], Keith Johnson [4], Jack L. Gallant [1,5] & Frédéric E. Theunissen [1,5,6] ✉

Speech processing requires extracting meaning from acoustic patterns using a set of intermediate representations based on a dynamic segmentation of the speech stream. Using whole brain mapping obtained in fMRI, we investigate the locus of cortical phonemic processing not only for single phonemes but also for short combinations made of diphones and triphones. We find that phonemic processing areas are much larger than previously described: they include not only the classical areas in the dorsal superior temporal gyrus but also a larger region in the lateral temporal cortex where diphone features are best represented. These identified phonemic regions overlap with the lexical retrieval region, but we show that short word retrieval is not sufficient to explain the observed responses to diphones. Behavioral studies have shown that phonemic processing and lexical retrieval are intertwined. Here, we also have identified candidate regions within the speech cortical network where this joint processing occurs.

The human cerebral cortex performs complex computations to transform a continuous speech sound pressure waveform into a language based message. These transformations include the detection of information-bearing spectro-temporal features in the speech sound, the combination of these acoustic features into phonemic units, syllables, words and meaningful clauses. At a coarse level, the location of cortical regions underlying the transformation from sound to words has been well described[1-3]. However, the neural basis of the phonemic processing steps where phonemic subunits are combined into meaningful larger units for the identification of syllables and words remains poorly understood. In order to explore phonemic processing in the cerebral cortex, we measured human fMRI blood-oxygen level-dependent (BOLD) response to narrative stories[4,5]. First, we contrasted the predictive power of linearized encoding models based on acoustic features, phonemic features and semantic features in order to delineate the cortical regions that were sensitive to phonemic information. Second, to explore the nature of phonemic segmentation within these identified phonemic cortical areas, we then contrasted the

predictive power of nested phonemic encoding models that used single phonemes, diphones and triphones as regressors.

Our approach implicitly combines two complementary analyses that have been used successfully in prior neurolinguistic research. The hierarchical processing in the sound to meaning transformation of the speech signal requires both abstraction at different levels and segmentation at different time scales. On the one hand, research based on fMRI experiments has principally leveraged differences in responses in predictive power for different levels of abstraction. For example, by using words versus non-word matched speech sounds as stimuli, researchers have analyzed BOLD responses to delineate cortical word-specific regions from phonemic regions within the temporal lobe[3,6-8]. Similarly, the predictive power of statistical linear models using spectral features versus articulatory features has been used to distinguish primary auditory cortical regions from speech specific cortical regions within the superior temporal gyrus[3,7].

On the other hand, research based on EEG, MEG and ECoG experiments have principally leveraged the entrainment observed at

[1]Helen Wills Neuroscience Institute, University of California, Berkeley, Berkeley 94720 CA, USA. [2]Departments of Neuroscience and Computer Science, University of Texas, Austin, Austin 78712 TX, USA. [3]Faculty of Electrical Engineering and Computer Science, Technische Universität Berlin, Berlin 10587 Berlin, Germany. [4]Department of Linguistics, University of California, Berkeley, Berkeley 94720 CA, USA. [5]Department of Psychology, University of California, Berkeley, Berkeley 94720 CA, USA. [6]Department of Integrative Biology, University of California, Berkeley, Berkeley 94720 CA, USA. ✉e-mail: lilyxuegong@berkeley.edu; theunissen@berkeley.edu

different temporal scales between the speech waveform and the neural signals measured with those techniques[9–11]. For example, neural correlates of word segmentation have been localized to cortical regions found along the left inferior and middle frontal gyri[12]. At lower frequencies (<2 Hz), it has also been shown that the time scales of phrase and sentence processing can be reflected in the neural activity found in middle and posterior superior temporal gyrus and inferior frontal gyrus[10]. At higher frequencies, corresponding to the syllable rate, the robust phenomenon of cortical entrainment to the speech envelope has been linked to lower level processing of speech sounds. It is well accepted, however, that the functional properties of this cortical speech entrainment as measured in EEG needs to be further analyzed in order to distinguish its role in speech or linguistic processing from a more general role in acoustic processing[13].

Here we assessed the predictive power of linear models based on phonemic units combined at different levels of *temporal* granularity and contrasted those models to those based on lower level (spectral) and higher level (word embeddings) of *abstraction*. This combined approach allowed us to clearly delineate phonemic cortical regions from acoustic or word/semantic regions and simultaneously examine the granularity of the segmentation for phonemic units. Although it appears counter-intuitive to be able to detect the fast rates of phonemic segmentation in the slow fMRI BOLD signal, we will show that this analysis is possible as long as individual voxels show sufficient sensitivity for the identity of specific phonetic units.

Lastly, the locus of the transition from phonemic processing to lexical retrieval has also been debated as both anterior and lateral posterior regions of the temporal lobe as well as inferior prefrontal cortex (reviewed in[14]). The investigation of the temporal granularity of phonemes in combination with the representation of speech

processing at different levels of abstractions allowed us to more rigorously assess the unique phonemic cortical region and thus revisit the locus of the phonemic to lexical retrieval transition. For this purpose, a detailed quantitative analysis of the differences in predictive power between linearized models based on phonemic features versus semantic features was performed.

## Results

The goals of this study were, first, to localize the phonemic brain regions; second, to investigate the nature of the phonemic segmentation; and third, to determine the putative brain regions where the transformation from phonemic processing to lexical semantic meaning representation occurs.

To achieve these goals, we collected fMRI BOLD data while 11 participants listened to more than two hours of spoken narrative stories from *The Moth Radio Hour*[3–5]. Following the logic of the voxelwise encoding model (VM) framework, acoustic, phonemic, and semantic features were first extracted from the speech stimuli (Fig. 1). These features were then used in regularized linear regression[15] to predict the time varying BOLD signals in each voxel independently. The predictive power of these voxelwise models was then estimated by cross-validation, using a separate data set reserved for this purpose (Methods, Supplementary Fig 1 and 2). By comparing the predictive power obtained using different feature spaces, one can obtain a functional map of phonemic processing.

### Responses to acoustic features and presence of speech sounds
In order to localize the cortical regions sensitive to phonemic processing, we first removed the fraction of the BOLD response that could be explained by the mere presence versus absence of speech sounds.

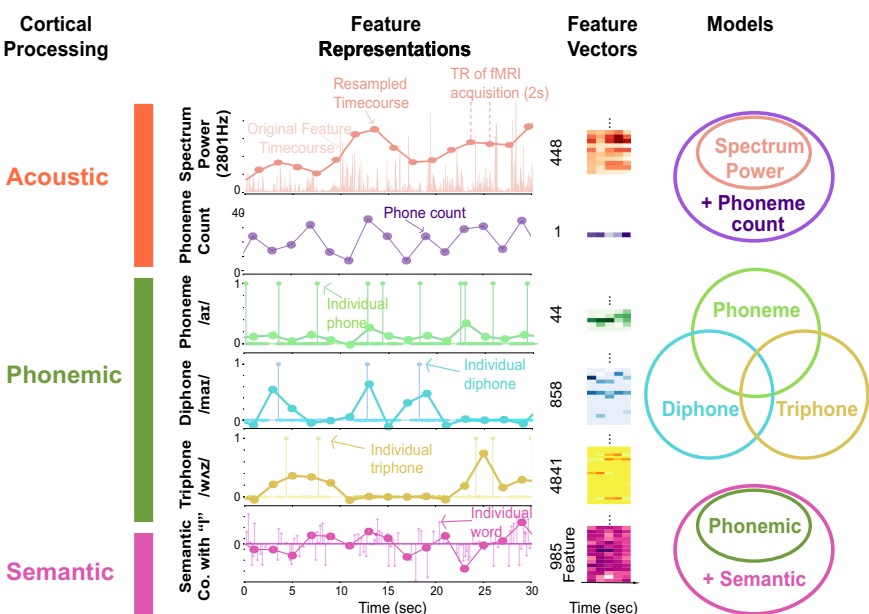

**Fig. 1 | Feature spaces used for voxelwise modeling.** To identify the cortical representation of features important for speech comprehension, the sound waveforms from the stimuli were first transformed into six different feature spaces. Acoustic features were extracted by creating separate spectrum power and phoneme count feature spaces. These two acoustic feature spaces describe brain activity that can be explained by the mere presence or absence of speech sounds. Phoneme-related features were extracted by creating separate feature spaces that reflect single phonemes, diphones and triphones. Semantic features were extracted by creating a feature space based on a 985-dimensional word embedding space[4]. Cortical Processing: Six distinct feature spaces were used to investigate the Acoustic, Phonemic and Semantic processing in the speech cortical network. Feature Representations: Illustration of the time course of a single feature from each feature space: the

spectrum of sound signal in the frequency band centered at 2801 Hz, the phoneme (/ai/), the diphone (/m.ai/), the triphone (/w.ah.s/), and the semantic co-occurrence with "I". These signals are low-pass filtered to generate the continuous values discretized at the TR (bold colored lines). Feature Vectors: Illustration of matrices of each feature space for 5 TRs and a subset of the features in each feature space. Models: Venn diagrams illustrating the features used in nested voxelwise models (VMs). The *Acoustic Baseline* VM used the spectrum power and the phoneme count as features. After subtracting the predictions from the *Acoustic Baseline* VM, nested models using phonemic and semantic features (green and pink circles at bottom) were fitted to localize phonemic regions and phonemic to semantic cortical boundaries. Nested models using single phonemes, diphones and triphones were fitted to assess the granularity of the phonemic segmentation in phonemic regions.

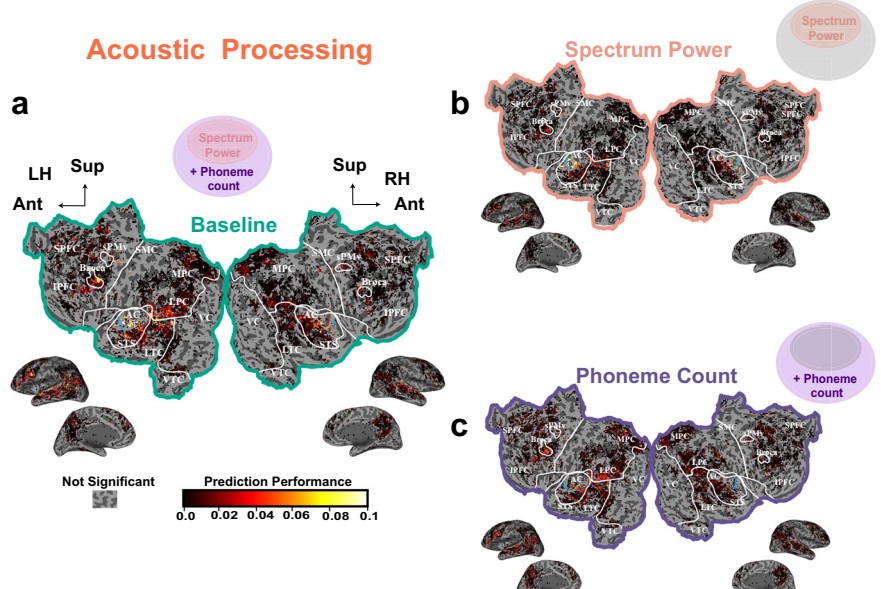

**Fig. 2 | Variance partitions for acoustic processing.** To identify acoustic processing across the cerebral cortex, a joint *Acoustic Baseline* VM consisting of spectrum power and phoneme count feature spaces was constructed. The prediction performance obtained from one typical subject (S5) for the joint *Acoustic Baseline* VM, and the models based on the time-varying spectrum power feature only, and the phoneme count feature only after excluding the time-varying spectrum power are shown. **a** Shows that the Acoustic Baseline VM produces accurate predictions of brain activity in PAC, LTC, sPMv and IPFC. To determine whether these representations were best modeled using spectrum power, or phoneme count features, variance partitioning was used to identify how much of the variance in brain activity could be explained by models based on each of these two features.

**b** Shows that spectrum power features best explain response variance in PAC. **c** Shows that phoneme count features best explain response variance in STG, sPMv and IPFC. Color shows the value of cross-validated prediction performance $R^2$ for all statistically significant voxels (Permutation test with FDR correction). White lines separate occipital, temporal, parietal and frontal cortex. White circles indicate regions of interest acquired from separate localizer scans. The cyan line within AC locates Heschl's gyrus. Anatomical regions labeled are: SPFC superior prefrontal cortex, IPFC inferior prefrontal cortex, MPC medial parietal cortex, LPC lateral parietal cortex, STS superior temporal sulcus, LTC lateral temporal cortex, MTC medial temporal cortex, VC visual cortex; Additional areas defined with localizers: AC auditory cortex, Broca, sPMv ventral speech premotor area.

For this purpose, we obtained the BOLD predictions from a joint model, *Acoustic Baseline* VM, consisting of the time-varying spectrum power and phoneme count features (Fig. 1). The phoneme count feature quantifies the presence versus absence of speech sounds without being linearly sensitive to the sound intensity or the exact spectrum of particular utterances. Within the *Acoustic Baseline* VM, we examined the anatomical location of voxels sensitive to time-varying spectrum power, and phoneme count individually (Fig. 2; see Supplementary Fig. 5 for per subject data). The time-varying spectrum power best predicts the BOLD response of voxels located in the most medial part of the auditory cortex (AC; identified functionally using localizers as described in the Methods). This medial region surrounds the Heschl's gyrus (cyan line in Fig. 2) where the primary auditory cortex (PAC) is found. Phoneme count best predicts the BOLD response of voxels in the more lateral regions of AC including voxels along the superior temporal gyrus (STG). Significant predictions are also observed in the more lateral ventral speech premotor area (sPMv) and inferior prefrontal cortex (IPFC) including Broca's area, mostly in the left hemisphere of this subject. These phoneme count auditory voxels could simply correspond to general auditory regions that are more sound level invariant than what can be predicted by linear weights on the time varying spectrum power. Alternatively, these phoneme count voxels could also correspond to an area of auditory processing specialized for speech. Since the goals of our study were not to functionally parcelate these auditory regions, we did not perform any additional analyses or experiments to disambiguate these two alternatives. In order to delineate the cortical regions sensitive to phonemic identity and not simply to their presence, we subtracted the predictions from this *Acoustic Baseline* VM from the BOLD response to obtain a BOLD response residual. All subsequent predictions refer to predictions for this residual response, which we will refer to as $Y_{res}$.

## Phonemic voxels and phonemic segmentation

To investigate phonemic processing, we defined phonemic voxels as the voxels that responded to the identity of speech phonemes and/or phoneme combinations beyond what could be explained by the mere presence or absence of speech sounds. The response to the identity of speech phonemes was assessed by significant predictions of the BOLD responses based on the full *Phonemic* VM. The full *Phonemic* VM consists of single phoneme, diphone and triphone features. Here we will first describe the anatomical location of the phonemic voxels. Second, we examine phonemic segmentation by quantifying the variance in brain activity that could be explained by models based on each of these three phoneme-related features and the joint of these features.

Anatomically, phonemic voxels were found in large regions of the temporal, parietal and prefrontal cortex (Fig. 3a: Phonemic Processing). In the temporal cortex, phonemic voxels were found in the superior temporal gyrus (STG) and had increasing predictive power in the superior temporal sulcus (STS) and the adjoining lateral area covering a large section of the lateral temporal cortex (LTC). These temporal cortex phonemic areas were found on both hemispheres and also had large anterior to posterior coverage (see Supplementary Fig. 7 and Supplementary Note for a detailed analysis of hemispheric differences). By contrast, predictive power was very low in the primary auditory cortical areas (PAC). The temporal phonemic area also extended more posteriorly into the lateral and posterior parietal cortex (LPC, PPC). Phonemic areas were also found bilaterally along the medial parietal cortex (MPC). Finally, the inferior and superior prefrontal cortex (IPFC, SPFC) contained clusters of phonemic voxels, including in areas just anterior and inferior to Broca's area and anterior to sPMv.

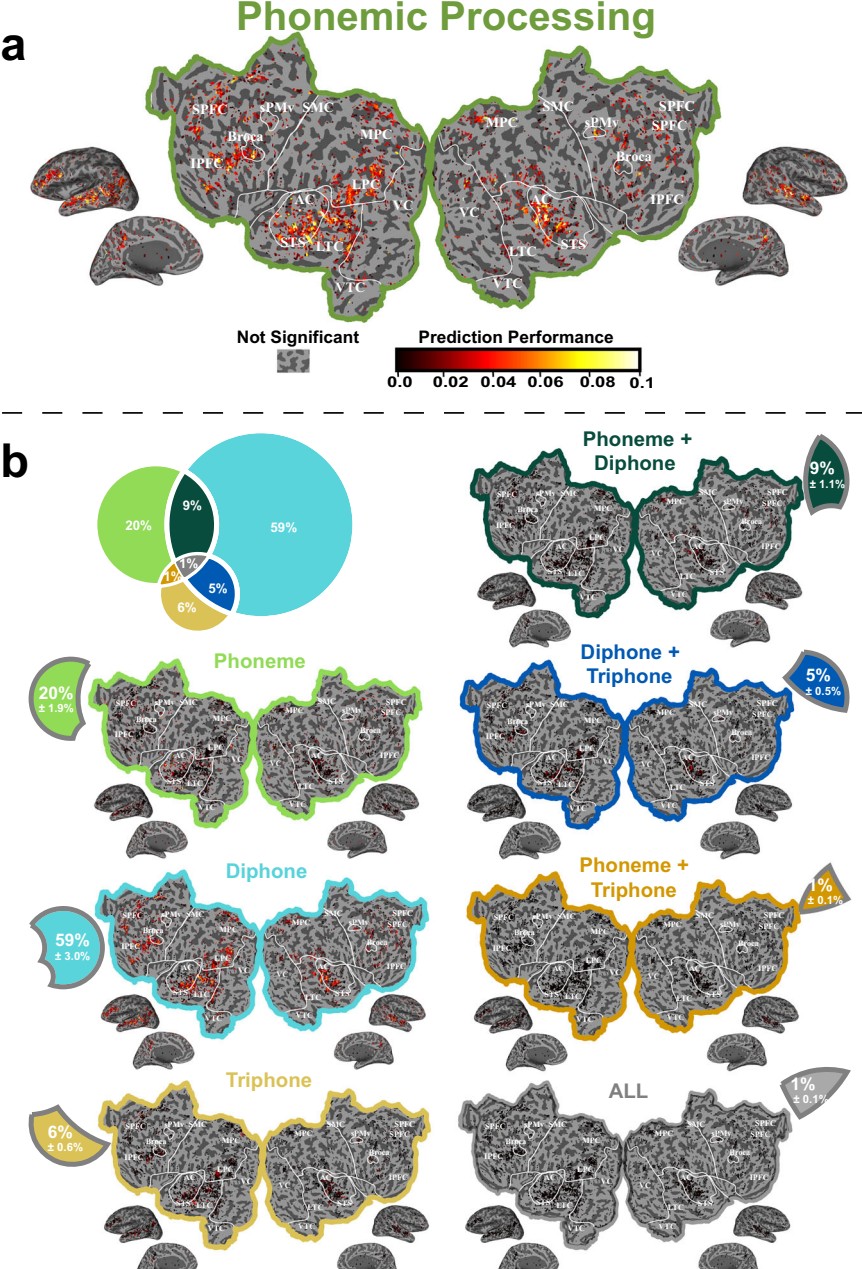

**Fig. 3 | Variance partitions for phonemic processing.** To identify phonemic representations across the cerebral cortex, a joint *Phonemic* VM consisting of single phoneme-, diphone- and triphone-based feature spaces was constructed. **a** Shows that the joint *Phonemic* VM produces accurate predictions of brain activity in LTC, LPC, MPC, IPFC and SPFC. To determine whether these representations were best modeled using single phoneme-, diphone-, or triphone-based features or the joint of these features, variance partitioning was used to identify how much of the variance in brain activity could be explained by models based on each of these three phoneme-related features and their joint pairs. **b** Shows that single phoneme features best explain response variance along STS. Diphone features best explain response variance in LTC, LPC, MPC, IPFC and SPFC. Triphone features and the joint of each pair of these phonemic features produce poor predictions in most voxels. Data used to generate this figure has been provided in source data. Cortical regions referred are: SPFC superior prefrontal cortex, IPFC inferior prefrontal cortex, MPC medial parietal cortex, LPC lateral parietal cortex, STS superior temporal sulcus, LTC lateral temporal cortex, MTC medial temporal cortex, VC visual cortex, sPMv ventral speech premotor area.

To visualize the unique and joint contribution of single phonemes, diphones and triphones to this cortical phonemic representation, we used variance partitioning. The single phoneme features alone significantly predicted the BOLD response of a small number of voxels located along the superior temporal sulcus (STS) of both hemispheres (Fig. 3b: Phoneme). By contrast, the diphone features (Fig. 3b: Diphone) significantly predicted the BOLD response in LTC, LPC, MPC, IPFC and SPFC. The unique contribution of the triphone features, and the joint contribution of each pair of

phonemic features and of all three phonemic features are relatively minor and scattered in voxels found in all phonemic cortical regions (Fig. 3b: Triphone, Phoneme + Diphone, Diphone + Triphone, Phoneme + Triphone, and All). Thus, cortical phonemic segmentation appears to occur principally at the diphone level. Single phoneme representation is limited to more "primary" speech processing areas along the STS. There is no clear distinctive triphone region beyond what was already identified as phonemic processing areas using the full *Phonemic* VM.

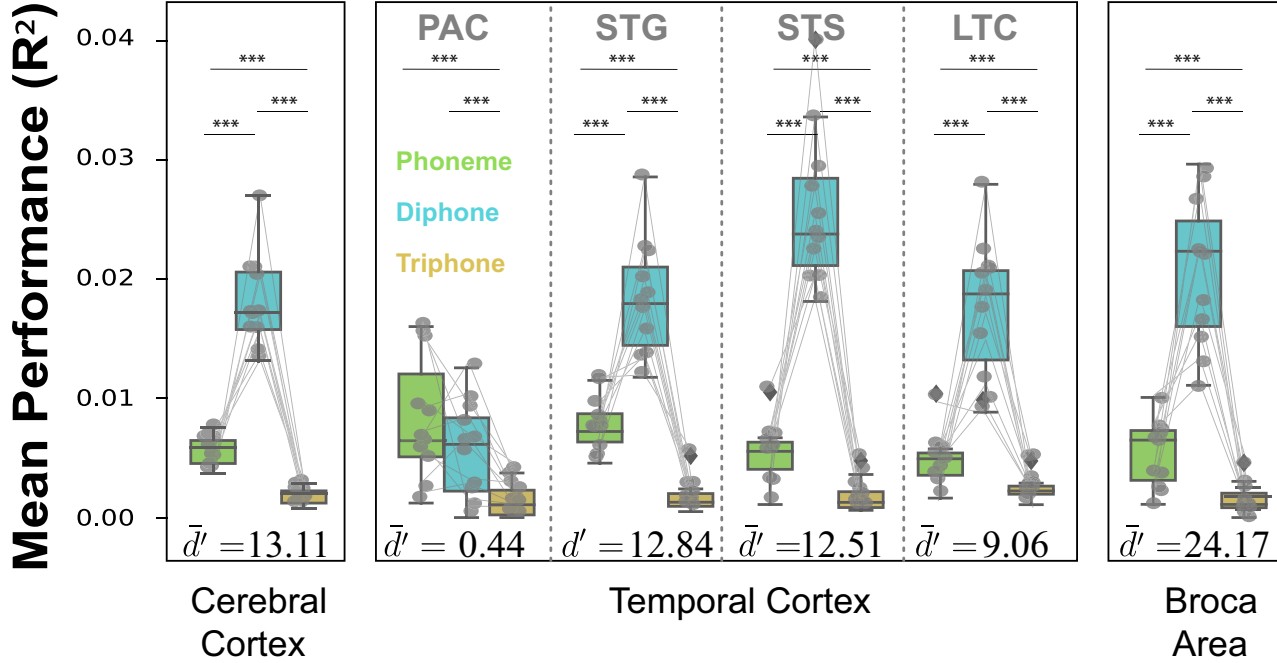

**Fig. 4 | Phonemic processing and segmentation.** To quantify phonemic representations across the cerebral cortex, the average prediction accuracy by ROIs (PAC primary auditory cortex, STG superior temporal gyrus, STS superior temporal sulcus, LTC lateral temporal cortex) for the unique contribution of single phonemes, diphones and triphones was shown in the box plots. It indicates that phonemic segmentation throughout the cortex, and specifically in regions in the temporal cortex and prefrontal cortex with high phonemic representation, occurs at the level of diphones, rather than single phonemes or triphones. The statistics are derived from the performance of significant phonemic voxels for each ROI ($n$ = 201 to 12,411 voxels) across 11 independent subjects (*$p < 0.05$, **$p < 0.01$ and ***$p < 0.001$). The effect size of difference is estimated by the average Cohen's d-prime ($\bar{d'}$) of diphone-single and diphone-triphone comparisons. All tests are two sided and corrected for multiple comparisons. The exact $p$-values can be found in the Phonemic voxels and phonemic segmentation section. For box plots, the horizontal lines show the first quartile, median, and the third quartile of the data. Whiskers extend to 1.5 times of interquartile range, and more extreme points are marked as outliers in diamonds. Data used to generate this figure has been provided in source data.

To quantify these observations, we estimated the unique contribution of each phonemic feature to the full *Phonemic* VM predictions in the entire cortex, in four ROIs partitioning the large phonemic areas found in the temporal cortex (PAC, STG, STS, LTC), and in Broca's area in the prefrontal cortex.

On average across the entire cortex, the unique contribution of the diphones is superior to the prediction obtained for the unique contribution of single phonemes and to the triphones (mixed-effect statistical models with subject as random factor: likelihood ratio test with null statistical model that ignores phonemic feature, $\chi^2(2) = 10355.60, p < 2.2 \times 10^{-16}$). The effect size is large: the averaged d-prime of the unique contribution of the diphone feature relative to the unique contribution of single phoneme and triphone features is $d' = 13.114 + - 0.203$ (2*SE*; obtained from mixed effect coefficients; Supplementary Table 1 and 2 for per subject data and analyses).

Within the temporal cortex (TC), the prediction performance of the full *Phonemic* VM shows a gradient: it is lowest in primary AC, highest in STG and STS, and lower again in LTC (Fig. 4). Furthermore, the diphone contribution varied relative to the single phoneme and triphone contributions as a function of anatomical location (Fig. 4). A linear mixed-effect statistical model analysis was used to predict the unique contribution to the prediction performance with the three phonemic features, the four ROIs in TC and their interaction as fixed effects and subject as random effect. This full statistical model has 12 coefficients (1 for intercept, 3 for ROIs, 2 for features and 6 for the interaction term). Prediction performance varied significantly across ROIs (likelihood ratio test with the nested statistical model that does not include ROIs and the interaction: $\chi^2(9) = 625.68, p < 2.2 \times 10^{-16}$) and across phonemic features (likelihood ratio test with the nested statistical model that does not include feature spaces and the interaction $\chi^2(8) = 4828.82, p < 2.2 \times 10^{-16}$). The unique contribution is higher for the diphone than for the single phoneme and triphone features in STG, STS, and LTC. Moreover, there was a significant interaction between cortical regions and feature space (likelihood ratio test with the nested statistical model that does not include the interaction $\chi^2(6) = 522.32, p < 2.2 \times 10^{-16}$). This interaction can be described by the effect size used to quantify the differences in the unique contribution between the diphone features and the single phoneme and triphone features in each ROI within the temporal cortex. The effect size is small in PAC ($\bar{d'} = 0.440 + - 0.968(2SE)$), increases in the STG ($\bar{d'} = 12.838 + - 0.838(2SE)$), and in the STS ($\bar{d'} = 12.509 + - 0.327(2SE)$), and then decreases in the LTC ($\bar{d'} = 9.060 + - 0.471(2SE)$). All $\bar{d'}$ and SE are obtained from mixed effect coefficients. Thus, in all phonemic regions of the temporal cortex, the unique contribution of the diphone features is the highest and the effect size of this difference is larger in STG and STS.

In Broca's area, the diphone features also significantly predict BOLD response better than single phoneme or triphone features (likelihood ratio test with the nested statistical model that does not include feature spaces, $\chi^2(2) = 443.64, p < 2.2 \times 10^{-16}$; Effect size is $\bar{d'} = 24.168 + - 0.203(2SE)$).

In summary, phonemic segmentation throughout the cortex, and specifically in regions in the temporal cortex and prefrontal cortex with high phonemic representation, occurs at the level of diphones, rather than single phonemes or triphones.

Finally, we also investigated to what extent these results could be affected by the fMRI methodology and, in particular, by a potential mismatch between the rate of individual phonemic units and the temporal resolution of the BOLD signal (Fig. 1). The rate of single phonemes in natural speech stimuli is rapid and varies from 10 to 40 phonemes per TR (i.e. up to 20 Hz). However, the rate of particular single phonemes, diphones and triphones is much slower. Thus, as long as a putative phonemic voxel is sufficiently sensitive for phonemic identity, one should be able to predict its BOLD signal from the presence and absence of a small set of particular phonemes in the voxel's "phonemic receptive field". The requirement of "sufficiently sensitive" depends on the signal to noise ratio of the BOLD signal, and on the length of the signal acquired during the experiment. To determine this sensitivity threshold, we performed a series of simulations where we systematically varied the sensitivity of model voxels (Methods). Using two hours of narrated speech data, the VM approach used on our data set would distinguish phonemic identity voxels from phonemic count voxels if the model voxel was sensitive for fewer than 10 particular single phonemes and fewer than 100 particular diphones (Supplementary Fig 3). The same simulations showed that one could also distinguish voxels that were sensitive to single phonemes from those that were sensitive to diphones or triphones but not to longer combinations (Supplementary Fig. 4).

## Diphone segmentation for identification, expectations or lexical retrieval?

Models capturing the diphone identity yielded, among phonemic features, the highest predictive power in large parts of the cortex implicated in speech processing. Does this phonemic segmentation at the level of the diphone reflect sensitivity for the identity of particular diphones or does it reflect other speech related processing such as expectations for specific diphone sequences or lexical retrieval?

A large body of previous work has shown the importance of predictions in speech processing[16–18]. In order to further explore if the cortical BOLD activities truly encode the content of diphones or merely the statistical properties of diphones, we built one diphone statistics model consisting of 8 phonological statistical features extracted from Irvine Phonotactic Online Dictionary (IPhoD;[19]). These features describe the phonotactic probability of diphones (Supplementary Table 6). As shown in supplementary Figs. 9 and 10, although transition probabilities across diphones yielded some predictive power in the phonemic regions, the prediction strength of VM models based solely on these transitions probabilities was much smaller than VM models that explicitly represented the diphone identity.

Furthermore, since many diphones are also short words (e.g. diphone "M.AY" could represent the word "my"), it is important to elucidate whether the unique explanatory power from diphone features could be a reflection of responses to high-frequency short words. In addition, the beginning of words are considered to be linguistically important in the process of lexical activation during spoken-word recognition[20]. In order to test these potential effects, we divided the diphone features into three categories (Fig. 5 upper panel): short words (e.g. "M.AY"), word beginnings (e.g "AE.N" in word "and"), and diphone residues not belonging to either previous groups (e.g. "N.D" in word "undid"). We then estimated the contribution to the prediction of a VM phonemic model based on diphones (but also including words made of single phonemes) coming from each of these three diphone categories. We then normalized these raw contribution values by the proportion of occurrences in each group to effectively obtain an average prediction measure per diphone for each category. As shown in Fig. 5, there is a significant difference among short words, word beginnings, and diphone residues in terms of their normalized contribution to the prediction of the diphone features (mixed-effect statistical models with subject as random factor: likelihood ratio test with the nested statistical model that does not include diphone

categories: $\chi^2(2) = 52.75$, $p = 3.5 \times 10^{-12}$). The diphone's dominance in explaining cortical BOLD can be in part, but not solely, explained by responses to short words. The contribution to the prediction of the short words is significantly higher than the beginning of words ($F(2,10) = 7.44$, $p = 6.6 \times 10^{-5}$) and diphone residuals ($F(2,10) = 8.62$, $p = 1.8 \times 10^{-5}$). It should also be noted that we did not observe distinct cortical subregions within cortical areas with phonemic representations where the responses to short words were systematically higher than those to word beginnings of other diphones (Supplementary Fig 8, Supplementary Table 3).

## Phonemic versus semantic cortical representations

The cortical areas with significant phonemic representations described above were large and overlapped significantly with cortical areas that have been previously assigned to lexical retrieval and semantic processing[3–5]. To determine whether these regions represent phonemic information, semantic information, or both, we compared prediction accuracy of *Phonemic* VM using single phoneme, diphone and triphone feature spaces to the accuracy of *Semantic* VM using semantic feature space (see Supplementary Fig 12 for the hemispheric analysis). The prediction accuracy of *Semantic* VM is obtained from variance partitioning from subtracting the prediction accuracy of *Phonemic* VM from that of *Phonemic-Semantic* VM using all three phonemic feature spaces and the semantic feature space (Fig. 1, Fig. 6a). As expected, phonemic and semantic cortical representations overlap (Fig. 6b, d). However, the relative contribution of the phonemic and semantic features can vary across cortical regions.

First, we quantified the phonemic and semantic features' contributions averaged across the entire cortex. These average values serve as a baseline to assess to what extent a particular cortical region of interest is more phonemic or more semantic. We found that the cortex is significantly more involved in semantic processing than phonemic processing (linear mixed-effect statistical model with subject as random effect and feature as a fixed effect;the likelihood ratio test with the nested statistical model that does not include feature spaces, $\chi^2(1) = 712.23$, $p < 2.2 \times 10^{-16}$). The effect size at the level of the whole cerebral cortex is large: the d-prime obtained from the differences in the additive contribution of the semantic feature and the predictive power of the phonemic feature is $d' = 2.018 + - 0.151(2SE)$ (see Supplementary Fig. 11, Supplementary Table 4 and 5 for per subject data and analyses).

Next, we examined the additive contribution of the semantic features in the four ROIs of the temporal cortex defined above. The predictive power of the *Phonemic-Semantic* VM is lowest in PAC, increases in STG and peaks in STS and LTC (Fig. 6c). Moreover, a linear mixed-effect statistical model with features (two levels: Phonemic and Semantic), ROIs (four levels: PAC, STG, STS, and LTC) and their interactions (3 coefficients) as fixed effects, subjects as the random effect, and the contributions to prediction performance as the response variable, shows that the prediction performance of semantic features is significantly different from that of the phonemic features across ROIs (likelihood ratio test with the nested statistical model that does not include ROIs and the interaction: $\chi^2(6) = 561.87$, $p < 2.2 \times 10^{-16}$) and across features (likelihood ratio test with the nested statistical model that does not include the interaction: $\chi^2(3) = 80.63$, $p < 2.2 \times 10^{-16}$). Phonemic features explain a significantly larger portion of response variance than semantic features in STG ($\chi^2(1) = 19.12$, $p = 8.1 \times 10^{-6}$) and STS ($\chi^2(1) = 51.75$, $p = 6.3 \times 10^{-13}$). In contrast, semantic features explain significantly more response variance than phonemic features in LTC ($\chi^2(1) = 36.65$, $p = 1.4 \times 10^{-9}$). These ROI analyses suggest a medial-lateral gradient of phonemic versus semantic representation within the temporal cortex. The most medial regions near the auditory cortex ($d = 0.155 + 0.513$) better represent non-phonemic acoustic features, more lateral voxels in STG ($d = 0.626 + 0.280$) and STS ($d = 0.617 + 0.171$) better represent phonemic features, and the most lateral voxels

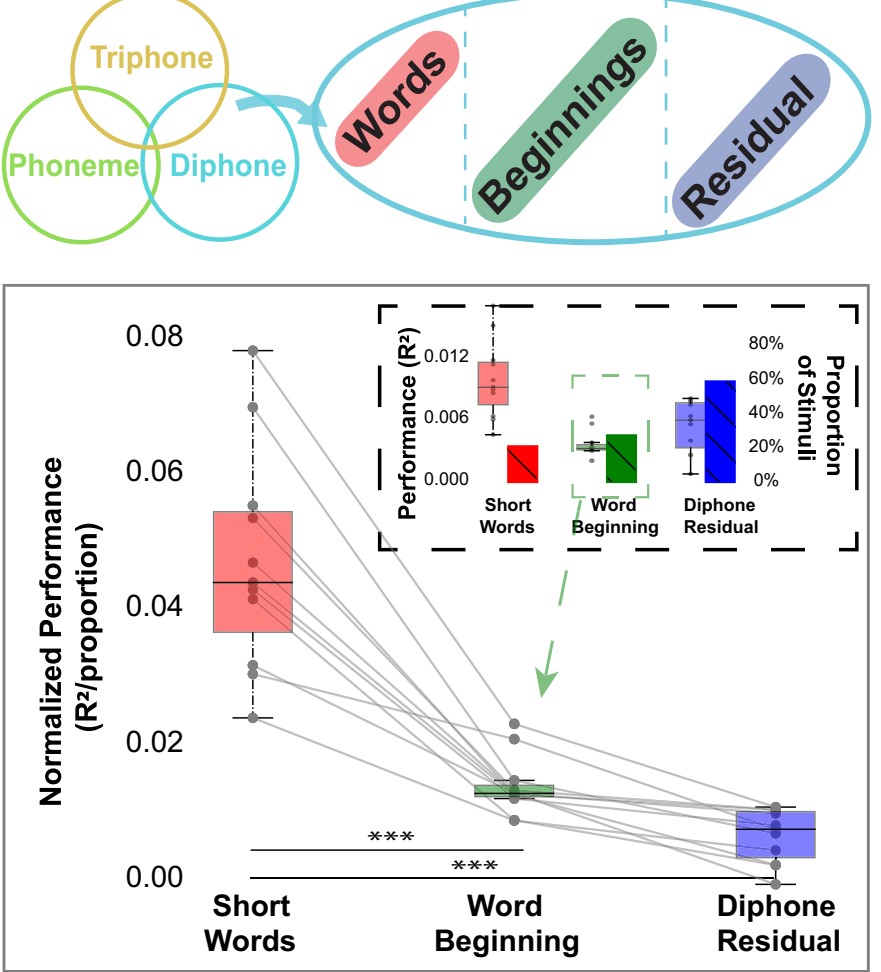

**Fig. 5 | Contribution of distinct diphone categories to phonemic VM.** In order to elucidate whether the unique explanatory power from diphone features could be a reflection of responses to high-frequency short words, top panel shows that diphones are split into three categories: short words, word beginnings and diphones that do not belong to these two categories called residual. Bottom panel shows that the average normalized prediction performance of short words is significantly higher than word beginnings and the diphone residual. The inset shows the fraction explained before normalization (lighter box) and the proportion of diphones in each group (solid bar). The statistics have been derived from all the significantly diphone voxels ($n$ = 15,396 to 16,001 voxels) examined across 11 independent subjects. All tests are two sided and corrected for multiple comparisons (*$p$ < 0.05, **$p$ < 0.01 and ***$p$ < 0.001). The exact $p$-values can be found in the Diphone segmentation for identification, expectations or lexical retrieval section. The box plots are defined the same way as in Fig. 4. Data used to generate this figure has been provided in source data.

in LTC ($d$ = 0.999 + 0.329) best represent semantic features. We will further delineate the boundary between the phonemic regions and the semantic regions in the last section of the results.

Within Broca's area in the prefrontal cortex, the additive contributions for the semantic features is significantly higher than the phonemic feature ($\chi^2(1)$ = 15.24, $p$ = 9.5 × 10⁻⁵). However, the effect size ($d'$ = 1.158 + − 0.593(2SE)) is smaller than the average observed across all cortical regions.

**Phoneme to semantic transitions**

The analyses described above revealed a gradient of phonemic to semantic cortical representation in the temporal cortex extending from its medial and superior region to its more lateral and posterior region. In addition, voxels in the inferior prefrontal cortex (IPFC) appear to be better predicted by phonemic features in their inferior portion and better by semantic features in their superior portion (Fig. 6, Supplementary Fig 13). We thus performed a more detailed anatomical analysis to examine the transition from phonemic to semantic representations in LTC and IPFC.

We projected the prediction performance of the *Phonemic* and *Semantic* VMs for all the significant voxels onto a "medial-lateral" axis

in LTC (perpendicular to STS), and "inferior-superior" axis in IPFC (perpendicular to IFS) (Fig. 7b, yellow lines). For each voxel in the corresponding region, we computed the distance to the nearest point along the bottom of the STS and the IFS for the LTC and the IPFC respectively. The average prediction performance along those two axes are shown in Fig. 7c: positive values are more medial (i.e. towards the STG) or more superior (i.e. towards the MFG), and negative values are more lateral (i.e. towards the MTG) or more inferior (i.e. towards the IFG). In LTC the contribution of the phonemic features to the overall prediction was highest around the STS (paired $t$-tests: −14 to 7 mm around STS of left hemisphere: $t(10)$ = 3.43, $p$ = 6.5 × 10⁻³; −12 to 20 mm around STS of right hemisphere: $t(10)$ = 5.36, $p$ = 3.2 × 10⁻⁴), while the additive contribution of the semantic features was higher around the Inferior Temporal Sulcus (ITS) (−30 mm lateral to STS of left hemisphere: $t(10)$ = −8.59, $p$ = 6.0 × 10⁻⁶; −40 mm lateral to STS of right hemisphere: $t(10)$ = −10.87, $p$ = 7.3 × 10⁻⁷). To coarsely quantify these effects, we estimated the physical location of the "center of mass" of the spatial density of the variance explained by phonemic versus semantic features for each subject (Supplementary Fig. 14). We found that in LTC, the center of mass of the phonemic feature contribution is consistently and significantly more medial than the center

**Fig. 6 | Phonemic to semantic processing.** To explore the cortical phonemic versus semantic processing, a joint *Phonemic-Semantic* VM consisting of single phoneme, diphone, triphone and semantic feature spaces was constructed. Variance partitioning was then used to obtain the cortical representation of phonemic and semantic models. The flat maps show the prediction performance for these models for one subject. **a** Shows that the *Phonemic-Semantic* VM produces accurate predictions in LTC, VTC, LPC, MPC, SPFC and IPFC. **b** Shows that the *Phonemic* VM produces accurate predictions in LTC, LPC, MPC, SPFC and IPFC. **d** Shows that the *Semantic* VM produces accurate predictions in VTC, LPC, MPC, SPFC and IPFC. **c** Show the mean and spread of the raw prediction performance across all subjects in ROIs in the temporal cortex, Broca's area and the entire cerebral cortex. It reveals that the *Phonemic* VM produces significantly higher prediction accuracy than the *Semantic* VM in STG and STS. In contrast, the *Semantic* VM produces significantly

higher prediction accuracy than the *Phonemic* VM in other regions of cortex, Broca's area and LTC (*$p < 0.05$, **$p < 0.01$ and ***$p < 0.001$). The statistics are derived from the performance of significant *Phonemic-Semantic* voxels for each ROI ($n = 2884$ to $380,229$ voxels) across 11 independent subjects. All tests are two sided and corrected for multiple comparisons. The exact *p*-values can be found in the Phonemic versus semantic cortical representations section. The box plots are defined the same way as in Fig. 4. Cortical regions referred are: PAC primary auditory cortex, STG superior temporal gyrus, STS superior temporal sulcus, LTC lateral temporal cortex, SPFC superior prefrontal cortex, IPFC inferior prefrontal cortex, MPC medial parietal cortex, LPC lateral parietal cortex, VTC ventral temporal cortex, MTC medial temporal cortex, VC visual cortex, sPMv ventral speech premotor area. Data used to generate this figure has been provided in source data.

of mass of semantic feature contribution (Supplementary Fig. 14, paired *t*-test: left hemisphere: $t(10) = 5.88, p = 1.1 \times 10^{-4}$, 11/11 subjects; right hemisphere: $t(10) = 9.28, p = 1.6 \times 10^{-6}$, 11/11 subjects). In IPFC, the phonemic contribution is significantly higher inferior to the IFS (−23 to −10 mm inferior to IFS of left hemisphere: $t(10) = 3.04, p = 6.5 \times 10^{-3}$; −27 to −23 mm inferior to IFS of right hemisphere: $t(10) = 3.58, p = 2.3 \times 10^{-2}$), while the additive contribution of the semantic features was higher superior to IFS in the left hemisphere only (20 to 30 mm superior to IFS: $t(10) = −4.70, p = 3.3 \times 10^{-3}$)(Fig. 7c). In summary, these analyses indicate that phonemic to semantic transition happens in a medial-lateral gradient in the LTC and an inferior-superior gradient in the IPFC.

## Discussion
We examined the anatomical location and granularity of phonemic representations in the human cerebral cortex. To clearly delineate the

phonemic cortical regions, we also examined the anatomical locations of acoustic and semantic representations and their overlap with the regions identified as being involved in phonemic processing. For these analyses, we trained voxelwise encoding models (VMs) to predict cortical fMRI signals elicited by natural narrative stories. The VMs used a hierarchy of feature spaces that reflect the transformation of speech sound to meanings: spectrum power, phoneme count, identity of phonemes, diphones, triphones and semantic embeddings.

We found phonemic cortical regions are primarily located not only in the superior and lateral temporal cortex (STG, STS, LTC), but also in the ventral and dorsal regions of the inferior parietal lobule (IPL) and in the inferior prefrontal cortex (IPFC). Within all phonemic areas, the segmentation at the level of the diphone yielded the most accurate predictions of the measured fMRI signal. The phonemic regions were distinct from primary auditory regions, but overlapped with cortical regions where word and semantic representations were found.

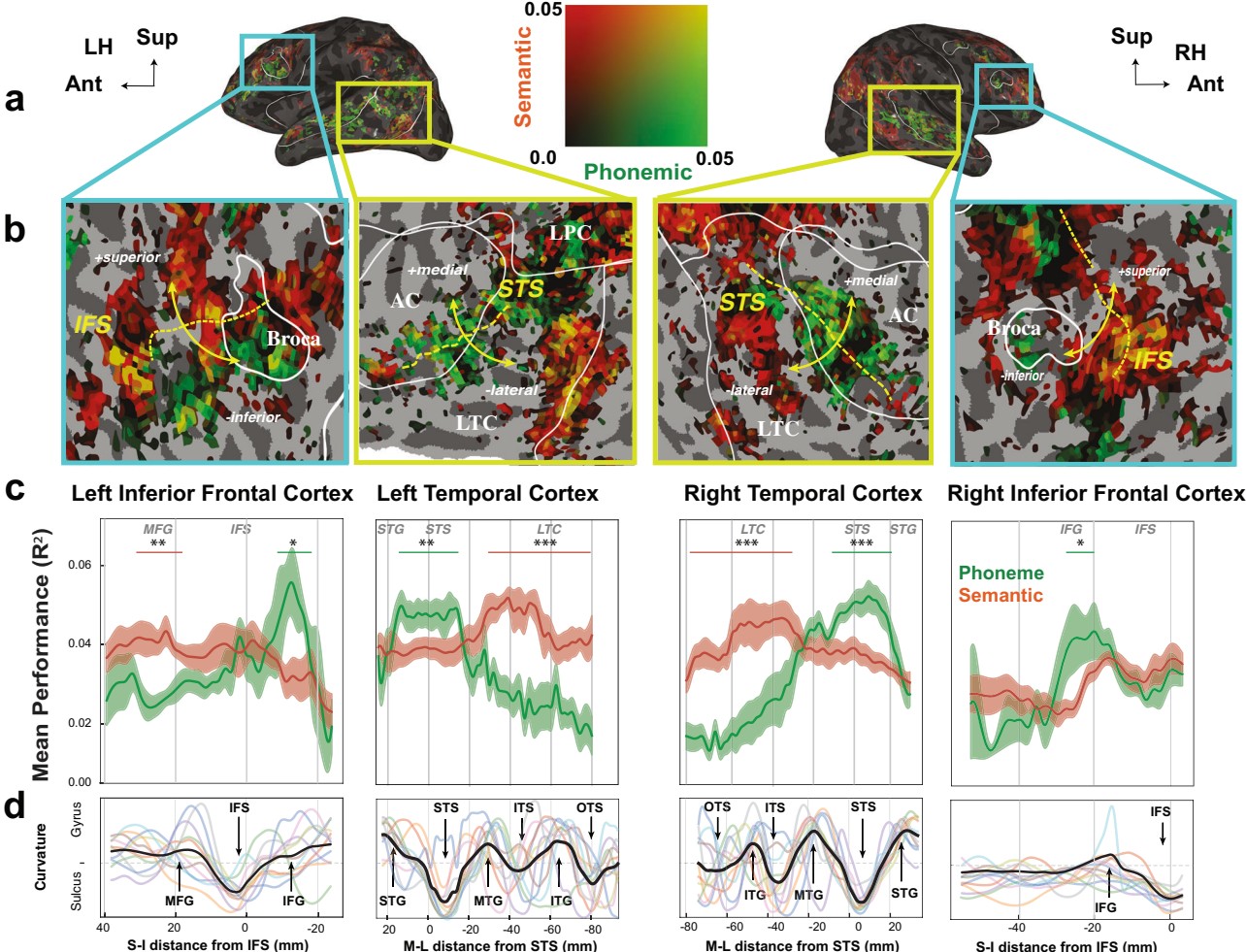

**Fig. 7 | Contribution to VM performance of phonemic vs semantic features in transition areas within the lateral temporal cortex (LTC) and inferior prefrontal cortex (IPFC).** To explore the gradient of semantic and phonemic representation in LTC and IPFC, voxels in LTC and IPFC were assigned a medial-lateral coordinate (M-L/S-I distance) based on their distance from the bottom of STS, and an inferior-superior coordinate (S-I distance) based on their distance from the bottom of IFS, respectively. The voxels were binned to 55 discrete bins in the LTC, and 30 bins in IPFC, aggregated across subjects. **a** Shows the performance of phonemic and semantic features on the lateral view of an inflated brain. **b** Shows the zoomed-in views of panel **a** in the superior temporal cortex and inferior prefrontal cortex. **c** Shows the average prediction performance across all subjects along those two axes. It reveals that the prediction accuracy of phonemic features is significantly higher around STS and inferior to IFS in both hemispheres, while the prediction accuracy of semantic features is significantly higher around ITS in both

hemispheres and MFG of left hemisphere (*$p < 0.05$, **$p < 0.01$ and ***$p < 0.001$). It suggests that phonemic to semantic transition happens in a medial-lateral gradient in the LTC and an inferior-superior gradient in the IPFC. Shaded areas show SE estimated across subjects. **d** Shows the average cortical curvature in each bin for each subject (colored lines) and aggregated across subjects (bold black line), with positive curvature indicating convexity (as on a gyrus) and negative curvature indicating concavity (as in a sulcus). All tests are two sided and corrected for multiple comparisons. The exact *p*-values can be found in the Phoneme to semantic transitions section. Cortical regions referred are: AC auditory cortex, STG superior temporal gyrus, STS superior temporal sulcus, LTC lateral temporal cortex, SPFC superior prefrontal cortex, IPFC inferior prefrontal cortex, MPC medial parietal cortex, LPC lateral parietal cortex, VTC ventral temporal cortex, MTC medial temporal cortex, VC visual cortex, sPMv ventral speech premotor area, IFS inferior frontal sulcus. Data used to generate this figure has been provided in source data.

Comparing the phonemic and semantic based predictions revealed a medial-lateral gradient in the temporal cortex: phonemic information was better represented in more medial regions (STG, STS), while semantic information was better represented in more lateral regions (LTC). An analogous inferior-superior gradient appeared in IPFC: phonemic information was better represented in more inferior regions of the IPFC, while semantic information was better represented in more superior regions.

One of the surprising results from this study is the large cortical areas in the lateral and posterior region of the temporal cortex involved in phonemic processing. In prior research, phonemic processing had primarily been assigned to small regions in the mid STG. In our prior work, we had used a phonemic feature space based on articulatory features describing single phonemes and found that the predictive power of this articulation based phoneme model was

limited to small regions in mid STG[3]. Similar conclusions were also reached in a meta-analysis of fMRI studies[1] as well as with ECoG recordings showing that electrodes placed on top of STG demonstrated tuning responses for groups of phonemes[21]. Recently, both intracranial recordings and stimulation studies indicated human mid STG might be a parallel higher level auditory area specialized for low-level speech processing[22]. Our study confirms these previous findings implicating the mid STG in phonemic processing: the VMs based on phoneme counts and the identity of single phonemes had the largest explanatory power in that region.

However, when diphones and triphones are added as distinct features in VMs, the cortical region in the temporal cortex implicated in phonemic processing is much larger. It extends laterally from STG to STS to much of the LTC. This large phonemic area in the LTC further extends into the ventral and posterior area of the IPL without a clear

break along the temporoparietal junction. Since diphones and triphones include many common short words, the phonemic regions defined here might also be cortical regions involved in lexical retrieval. In another word, these regions could be involved in the segmentation of the speech signals into known words. In order to disambiguate lexical processing from phonemic processing, we estimated the contribution short words made to our VM (Fig. 5). Indeed, we found that approximately half of the explanatory power originating from the diphone features could be attributed to short words. In addition, the semantic feature space based on word embedding is by necessity correlated with a lexical space. We showed that there was a large number of phonemic voxels where the additional predictive power obtained from the semantic feature space was significant. Thus, we propose that the large cortical regions where the Phonemic VM yielded good predictions of the BOLD response are cortical areas involved both in phonemic and lexical processing.

In the more lateral regions of the temporal cortex, the semantic features yielded additional predictive power that was greater in magnitude than the one obtained by phonemic features only, suggesting that this lateral region constitutes a cortical network implicated in the phonemic/lexical to semantic transition. Both this network in the lateral temporal cortex[23] and the adjacent region in the IPL[24] have been implicated in lexical retrieval and speech comprehension in multiple studies. We also found discrete locations within IPFC around the IFS where the phonemic model yields better predictions than the semantic model. These locations might also be involved in lexical retrieval potentially under a different linguistic context than the one found in lateral temporal cortex[25]. For example, it has been suggested that these regions in the IPFC are implicated in lexical retrieval with higher linguistic load such as with increase word cohort competition[26].

One key finding from this study is that, within the phonemic/lexical regions in the LTC, IPL and IPFC, segmentation occurs principally at the level of the diphone. This finding is consistent with basic phonetic theory. In natural speech, single phonemes don't occur in isolation and it is well known that single phonemes are acoustically different when produced in combinations, a phenomenon known as coarticulation[27]. In other words, the acoustic features characterizing single phonemes vary depending on the phonological context. This acoustic variability prevents a simple transformation from temporal-spectral acoustical features to the identification of a single phoneme. This acoustical variability is reduced when segmentation occurs at the level of diphones. Since the number of diphones used in languages remains relatively small (e.g. around 860 in English) compared to memory capacity of the human brain, it is reasonable to postulate that the transformation between an acoustic based neural representation and a phonemic based neural representation of a speech stream could therefore rely principally on a direct operation on spectral-temporal features to extract the diphone identity. As caveats, first, we have not exhaustively explored segmentation based on other speech features. For examples a segmentation based on the speech signal envelope (e.g. syllables, envelope peaks, etc) have shown to have large explanatory power for recovering the neural signals in the STG obtained in ECoG recordings[28]. Second, we have only investigated models with a static segmentation. More sophisticated models could also include an adaptive segmentation model based both on acoustic features and linguistic features as proposed in the adaptive resonance theory[29]. Third, the dominance of diphones may also reflect the engagement of predictive speech processing mechanisms which become more important in noisy situations[30]. In particular, under the noisy fMRI scanning environment, diphones could be more reliably represented and resistant to noise than single phonemes. On the other hand, the predictive power of VM that were solely based on the statistical properties of diphone sequences and ignored diphone identities was significantly smaller

(see Supplementary Fig. 9 and 10). Finally, our conclusions are based on measures of predictions of nested models of phonemic features. Based on a systematic variance partitioning of single phonemes versus diphones and triphones, we are confident that the unique variance explained by diphones over single phonemes is substantial while the unique variance explained by triphones over diphones is small. However, we don't provide direct evidence for diphone specificity, for example in the form of finding a cortical map that is systematically organized along diphone structure but not triphone structure. Additional explorations on the exact nature of the cortical representation for phonemic segments is needed.

As mentioned above, the over-representation of diphones in phonemic brain regions might also be related to word segmentation, a key process in lexical retrieval. Many common short words are diphones. To begin to assess the extent of putative word processing within the regions identified here as phonemic, we sorted diphones into three groups: short words, word beginnings and other. We found that the diphones that were short words were better represented than word beginnings and the other diphones. This over-representation might thus reflect the role of STG, STS and/or LTC in word segmentation and lexical retrieval. However, within the phonemic/lexical regions, we did not find subregions that were principally phonemic or principally lexical. These results are consistent with lexical retrieval mechanisms as revealed and established in behavioral studies that the process of word recognition entails gradual integration across multiple phonemes[31,32]. Thus phonemic and lexical processing appear to be intertwined both mechanistically and in cortical functional mapping.

Finally, we also assessed the extent with which our findings on the granularity of phonemic segmentation could be a consequence of the spatial and temporal resolution of the fMRI signal. The low temporal resolution of fMRI due to sluggish hemodynamic response function (HRF) signal has long been considered as a challenge to study fast brain processing like speech segmentation[33]. Our simulations (Methods), however, show that the capability of detecting fast neural activities underlying speech processing using fMRI is not necessarily constrained by this low temporal resolution as long as the fast speech events being studied are sufficiently variable at the temporal scale captured by fMRI. For example, although the phoneme counts in speech are high, in a given TR there will be a small number of counts of any particular phoneme or diphone, etc. Thus, in order to have temporally sparse events in the BOLD signal, the sensitivity of a single voxel must be sufficiently high as to only be responsive to a small number of phonemic units. The threshold of sensitivity is determined by the signal to noise ratio (SNR) and the data size. It is also a function of the spatial resolution of the fMRI[34,35].

Another key finding from this study is that two principal regions where the phonemic/lexical representation to semantic meaning transition occurs were identified: a medial-lateral gradient in the LTC and an inferior-superior gradient in the IPFC around IFS. In the LTC, the phonemic/lexical representation dominates in the more medial parts around the STS, while the semantic representation dominates in the more lateral areas along ITS. In our study, the phonemic/lexical LTC area includes the posterior medial temporal gyrus (pMTG) and the posterior part of the inferior temporal sulcus (pITS). These regions have been assigned as a core phonological to lexical interface in a large body of research based on clinical, neurophysiological and imaging research (reviewed in[14]). While our results and conclusions are consistent with that published work, we also found that the posterior LTC extends into the ventral and posterior area of the IPL making a larger continuous region of phonemic/lexical representation to semantics transition. On the one hand, the junction of the temporal and the parietal cortex (in particular the regions that include the supramarginal gyrus and angular gyrus, that together are known as Wernicke's area) has been extensively implied in speech processing. For example, it is known to be involved in categorical phoneme perception[36] and

more generally in auditory to motor integration[37]. On the other hand, the posterior IPL had not been identified previously as a region involved in the transition from phonemic/lexical representations to semantics. It has, however, been identified as a core region where visual representations are transformed into visual objects with semantic meaning[38]. Similarly, it is a region of transition between a visual representation of the letters in written text to the meaning of the words in the text[5]. The ventral and posterior IPL therefore appears to be implicated in the extraction of meaning from sensory stimulus irrespective of the modality. For the speech sound to semantic meaning transformation, it remains to be seen whether the ventral and posterior IPL serves a different function as the neighboring region in the temporal cortex. For example, one could hypothesize that the subregion in the temporal cortex specializes in the phonemic/lexical to meaning transformation supported by linguistic priors, and that the subregion in the parietal cortex specializes in the phonemic/lexical to meaning transformation by engaging the visual imagery that can be elicited by listening to narrated speech stories.

In the inferior frontal cortex, we found an inferior-superior gradient with a dominant phonemic/lexical representation around IFS and a dominant representation for semantics along the MFG. This finding is also in accordance with prior studies. IPFC is known to play a significant role not only in speech related sensori-motor integration but also in phonological processing, in particular when it involves analyses requiring segmentation based on lexical features[26,39]. In non-humates primates, high-level auditory neurons in IPFC have also been shown to respond categorically to species specific vocalizations according to their behavioral meaning[40].

In summary, by investigating phonemic segmentation in long fMRI recording sessions obtained while subjects listened to engaging narrative speech, two large regions involved in the representation of phonemic and lexical features were described: one in the superior and lateral temporal cortex and ventral/dorsal parietal cortex and a second one in the IPFC. In both regions phonemic features beyond those found in short words increase predictive power of encoding models. It indicates phonemic and lexical processing may be intertwined. In both LTC and IPFC, a systematic gradient in the relative dominance in phonemic/lexical versus semantic representation was also observed. The actual potential distinct roles of these two large cortical networks, or of subnetworks within these two large networks, in phonemic processing and in the phonemic/lexical to semantic transformation remain poorly understood. We also demonstrated here that fMRI analyses could be leveraged to study cortical speech processing, even those occurring at time scales that are faster than the temporal resolution of the BOLD signal. Thus, further exploring the predictive power of more complex VMs applied to fMRI data could allow for a finer functional parcellation of phonemic/lexical specific areas. For example, to gain further insights on networks dedicated to word segmentation, one could test the additional predictive power of phonemic based VM that also take into account word segmentation expectations based on linguistic context assessed at multiple time-scales[10,16,26,41,42]. Similarly, distinct roles for the subnetworks involved in phonemic/lexical to semantic transitions could also be tested. For example, linguistic priors that play a role in predicting meaning from the beginning of sentences could also be included as features in semantic based VMs[43]. Lastly, VMs could also be used to explore the speech representations of bilingual or multilingual subjects listening to stories in multiple languages including some that subjects do not understand. Such experiments would allow researchers to further distinguish phonemic segmentation and identification that occurs independently of understanding, or that might be common across multiple languages. These analyses would therefore be useful to also more clearly delineate the purely phonemic regions from those implicated in lexical retrieval, if such regions exist.

## Methods

### Participants
Structural and functional brain data were collected from seven male subjects (S1: age 26, S2: age 31, S5: age 30, S6: age 25, S7: age 36, S9: age 24, S10: age 24), and four female subjects (S3: age 28, S4: age 25, S8: age 24, S11: 31). Each subject's handedness was evaluated by the Edinburgh handedness inventory[44]. All subjects were healthy and had no reported hearing problems. The use of human subjects in this study was approved by the UC Berkeley Committee for the Protection of Human Subjects. A written statement of informed consent has been obtained from each subject.

### Stimuli
The stimuli in these experiments were pre-recorded stories from the Public Radio Exchange (PRX) radio show "The Moth Radio Hour", which has been used in previous studies of our lab[3–5]. These sounds have been annotated for their word and phonetic content. They are engaging stories that capture the attention of the subjects. The scanning sessions lasted approximately 2.5 h and included time needed to play the audio/visual stimuli used as generic localizers and two one-hour sessions during which the subjects listened to Moth stories.

The stimuli were split into separate model estimation and model validation sets. The model estimation stimulus-set consisted of ten 10- to 15-min stories played once each. The length of each scan was tailored to the story and also included 10 s of silence both before and after the story. Each subject heard the same 10 stories, 5 of which were told by male speakers and 5 by female speakers. The model validation stimulus-set consisted of a single 10-min story told by a female speaker that was played twice for each subject in order to estimate voxel response reliability and noise ceiling. This resulted in 3737 time points (sampled at TR) for the training dataset and 291 time points for the validation dataset.

Auditory stimuli were played over Sensimetrics S14 in-ear piezo-electric headphones (Sensimetrics MA, USA). These headphones provide both high audio fidelity and some attenuation of scanner noise. A Berhinger Ultra-Curve Pro hardware parametric equalizer was used to flatten the frequency response of the headphones (Berhinger, Las Vegas, USA). The sampling rate of the stimuli in their digitized form was 44.1 kHz and the sounds were not filtered before presentation. Thus, the potential frequency bandwidth of the sound stimuli was limited by the frequency response of the headphones from 100 Hz to 10 kHz. The sounds were presented at comfortable listening levels. Stories were manually transcribed and converted into separate word and phoneme representations (see[3,4]).

### MRI data collection and preprocessing
Structural MRI data and blood oxygen level dependent (BOLD) fMRI responses from each subject were obtained while they listened to approximately 2 h and 20 min of natural stories. For nine of the subjects, these data were collected during two separate scanning sessions that lasted no more than 2 h each. For two of the subjects (S1 and S5) the validation data (two repetitions of a single story) were collected in a third, separate session. MRI data were collected on a 3T Siemens TIM Trio scanner at the UC Berkeley Brain Imaging Center, using a 32-channel Siemens volume coil. Functional scans were collected using a gradient echo-EPI sequence with repetition time (TR) = 2.0045 s, echo time (TE) = 31 ms, flip angle = 70 degrees, voxel size = 2.24 × 2.24 × 4.1 mm, matrix size = 100 × 100, and field of view = 224 × 224 mm. 32 axial slices were prescribed to cover the entire cortex. A custom-modified bipolar water excitation radiofrequency (RF) pulse was used to avoid signals from fat tissue. Anatomical data were collected using a T1-weighted MP-RAGE[45] sequence on the same 3T scanner.

Each functional run was motion-corrected using the FMRIB Linear Image Registration Tool (FLIRT) from FSL 4.2[46]. All volumes in the run were then averaged to obtain a high quality template volume. FLIRT

was also used to automatically align the template volume for each run to the overall template, which was chosen to be the template for the first functional run for each subject. The overall template is obtained within each participant over all runs. These automatic alignments were manually checked and adjusted for accuracy. The cross-run transformation matrix was then concatenated to the motion-correction transformation matrices obtained using MCFLIRT, and the concatenated transformation was used to resample the original data directly into the overall template space. This overall template space is also obtained within each participant across all runs. Low-frequency voxel response drift was identified using a 2nd order Savitsky-Golay filter with a 120-s window, and this was subtracted from the signal. After removing this time-varying mean, the response was scaled to have unit variance (i.e. z-scored). Structural and functional MRI were combined to generate functional anatomical maps that included localizers for known regions of interests (ROIs). The method used to determine these ROIs is explained in detail in[3].

### Feature space and model construction

In order to localize the phonemic brain regions, to investigate phonemic segmentation and the phoneme to word/meaning transitions, we constructed six distinct feature spaces: time-varying power spectrum, phoneme count, single phoneme, diphone, triphone, and semantic features.

The time-varying power spectrum was obtained by estimating the power for 2-s segments of the sound signal between 25 Hz and 15 kHz, in 33.5 Hz bands (number of frequency bands = 448), using the classic Welch method for spectral estimation density as we have done previously[3].

The phonemes rate features consisted of phonemes' counts per TR. The time-varying power spectrum and the number of phonemes were combined to make the baseline feature space (number of features = 448 + 1 = 449). The baseline feature space quantifies the presence or absence of sounds and speech sounds. As we were interested in brain regions that process phoneme identity and meaning, the fraction of the BOLD response explained by the baseline feature space was analyzed separately and then subtracted.

The single phoneme feature space is composed of a binary vector encoding the presence or absence of 15 consonants and 24 vowels of English that we generated based on the CMU pronouncing dictionary[47]. Although the numbers of all possible diphones ($39^2 = 1521$) and triphones ($44^3 = 59319$) are large, English words utilize only a subset of all possible combinations. For the stories used in this study, we found 835 diphones and 2677 triphones. When we added combinations found in the larger IPhOD data set, we obtained altogether 858 possible diphones and 4841 possible tri-phones[19]. For diphone and triphone feature space, they are composed of one-hot code encoding the presence or absence of all possible diphones or triphones. In semantic feature space, each word is represented as a 985-dimensional vector based on word co-occurrence statistics with the 985 basis words from Wikipedia's "List of 1000 Basic Words" ([3,38]). All the feature vectors were downsampled to the TR using Lanczos filtering (Fig. 1).

### Voxelwise model fitting and validation

Based on these feature spaces, we created different linear encoding models (linearized regression) in order to predict the time-varying BOLD response of each voxel of each subject from the time varying stimulus (Fig. 1). We have called this approach voxelwise modeling (VM). First, we constructed a baseline VM that used spectrum power and phoneme count as regressors ($X_B \in R^{nt \times 449}$) to predict the BOLD response ($Y \in R^{nv \times nt}$), where nt is the number of time points ($nt = 3737$) and nv is the number of voxels (varies across subjects; range: 73, 023 – 92, 970). We then subtracted this prediction from the measured BOLD response to obtain a response residual ($Y_{res} \in R^{nv \times nt}$) before

fitting VMs based on phonemic and semantic identities. Subtracting the prediction from this baseline VM is needed in order to distinguish variance in the BOLD response that is simply due to the presence versus absence of phonemes or words from the variance that is dependent on the identity of phonemic and semantic features. Second, one joint phonemic model, using single phonemes, diphones and triphones features ($X_{1+2+3} \in R^{nt \times 5738}$) as regressors, was fitted to predict the BOLD response residual, $Y_{res}$. Finally, the phoneme-semantic VM was obtained by using all three phonemic features with semantic features ($X_{1+2+3+4} \in R^{nt \times 6723}$). Before fitting the VM weights (also known as model coefficients in the context of linear regression), the regressors X and the BOLD response Y were z-scored. BOLD response was z-scored separately for each story to control for the random effect of the story.

In order to account for the temporal integration time constants of both the neural processing and the BOLD response, our linear models predict the BOLD response at time t from signal features evaluated in four time windows of 2 s each and starting at $t - 2$ s, $t - 4$ s, $t - 6$ s and $t - 8$ s. This is accomplished by concatenating feature vectors that had been delayed by 2 s, 4 s, 6 s, and 8 s. This yields 1796 features (448 spectrum power × 4 + 1 phoneme count × 4) for the *Acoustic Baseline* VM, 22,952 features (39 phonemes × 4 + 858 diphones × 4 + 4841 triphones × 4) for the *Phonemic* VM, and 26,892 features (39 phonemes × 4 + 858 diphones × 4 + 4841 triphones × 4 + 985 words × 4) for the *Phonemic-Semantic* VM.

Due to the relatively large number of features relative to *nt*, we used regularized regression techniques to estimate the VM weights to prevent overfitting. Regularization was achieved using Tikhonov regression with different levels of regularization used for different feature spaces and delays for each voxel and each subject. Along the spatial dimension (features), a different regularization hyperparameter was used for each feature space. For example, all features belonging to the single phoneme feature space would shrink according to $\lambda_1$ (the regularization hyperparameter corresponding to the inverse variance of the Gaussian prior), while all features belonging to the diphones would shrink according to $\lambda_2$[3,15,48]. We have called this type of regularization banded ridge regression because the diagonal covariance matrix that characterized the Gaussian prior has bands corresponding to the equal values of for all features belonging to the same feature space. In addition to the regularization along the spatial dimension, our regularization procedure also acted along the temporal dimension (the features in the four delay periods). As mentioned in the previous paragraph, the weights along the temporal dimension include the neural integration time and the time course of the hemodynamic response. As part of the regularization procedure, one can capture the expected effect due to the hemodynamic response by using a hemodynamic response function (HRF) temporal prior. This temporal prior was fixed and effectively enforces different levels of regularization for different time delays. An efficient analytical solution for banded ridge regression was implemented using the kernel method as described in[15,48].

For each feature space and each voxel, we tested 10 possible regularization hyperparameters, log spaced between $10^2$ hyperparameters for the baseline VM, $10^3$ for the phonemic VM and $10^4$ for the phoneme-semantic VM. This hyperparameter search was performed for each voxel of each subject. The optimal hyperparameter was found by maximizing the cross-validated coefficient of determination ($R^2$) using a 10 fold cross-validation procedure on the training data set (see below). In order to prevent overfitting, these optimal hyperparameters were tested on a separate validation data set.

Finally to quantify the goodness of fit of the VMs, the fitted VMs were used to predict BOLD responses to a separate story that had not been used for hyperparameter optimization mentioned above and model fitting. Prediction performance was then estimated from the $R^2$ calculated between predicted and actual BOLD responses for each voxel over the 291 time points for this validation data.

## Variance partitioning

In order to quantify the variance and localize the cortical regions uniquely explained by individual feature spaces and any combination of feature spaces, we performed a variance partitioning procedure. We first fitted a joint phonemic model with all phonemic features (single phonemes, diphones and triphones)[3]. Then, we used the hyperparameters (to the optimal shrinkage) obtained from this joint phonemic model of each block of features in the banded regression so that the same sub space of features obtained by the ridge is used in all models. We computed the variance of seven possible partitions: the unique variance of each feature and all the possible combinations of these features (single phonemes + diphones, diphones + triphones, single phonemes + triphone, single phonemes + diphones + triphones). Afterwards, we performed a variance correction on $R^2$ that is used to eliminate biases in the $R^2$ estimation that could lead to nonsensical results.

In a separate analysis, we examined the contribution of distinct types of diphones features. First, we separated diphone into three categories: diphones that are short words (e.g. "M.AY" as word "my"), diphones that are at the beginning of words (e.g. "AE.N" in word "and"), and all the other diphones. We then estimated the contribution of each diphone category to the prediction by deleting the diphones that did not belong to the category being considered; more precisely we set the contribution to the diphone feature vector from the out-of-considered-category diphones to zero. With this procedure, we divided the predicted signal obtained from the diphones into three components, one obtained for each mutually exclusive diphone category.

## Definition of ROIs

Regions of interests including auditory cortex broadly defined (AC), Broca's area, and ventral speech premotor area (sPMv) were defined based on standard functional localizer scans. Voxels that were responsive when the subject listened to 10 repetitions of a one-min auditory stimulus with 20-s segments of music, speech and natural sound are considered to belong to AC. AC thus includes both primary and secondary auditory cortical areas. When the subject continuously subvocalizes self-generated sentences, the active voxels located at the triangular part of the inferior frontal gyrus are determined as Broca's area and voxels located at the premotor cortex as sPMv. The repeatability of the voxels' response was calculated as an F statistic given by the ratio of the total variance responses over the residual variance.

Other ROIs were defined based on speech processing relevant anatomical landmarks[4,49–52] obtained from the structural MRI scan. In particular, the inferior prefrontal cortex (IPFC) contains cortical regions ventral to inferior frontal sulcus (IFS), while superior prefrontal cortex (SPFC) contains regions dorsal to superior frontal sulcus (SFS). The peak of the superior parietal gyrus separates the lateral parietal cortex (LPC) from medial parietal cortex (MPC). LPC contains supramarginal gyrus (SMG) and angular gyrus (AG). The peak of the inferior temporal gyrus is used to separate the lateral temporal cortex (LTC) from ventral temporal cortex (VTC).

## Statistical analyses

The statistical significance of the cross-validated coefficient of determination $R^2$ was estimated using a permutation analysis. First, We obtained a set of regularization hyperparameters from fitting the phoneme-semantic VM (using single phoneme, diphone, triphone and semantic features) for each subject. Then, a permuted $R^2$ for each model of each subject was obtained by refitting the model using this set of hyperparameters and the shuffled regressors of each feature space within the model. This process was repeated 1000 times to generate the null distribution of cross-validated $R^2$ for each voxel of each subject. Based on the values of this null distribution, we used variable thresholds yielding a significance level of 1% corrected using

the False Discovery Rate (FDR) procedure[53] to determine statistical significance.

The raw $R^2$ values, or the partial $R^2$ raw values obtained from variance partitioning, obtained for each voxel and each subject were then aggregated to make inferences about reliable patterns of correlations between features in the speech stimuli and BOLD activity in specific brain regions. Alternatively, we also used a winner-take-all approach assigning the feature space that yielded the highest prediction in a specific comparison (e.g. single phoneme, diphone, triphone) as a label to each voxel. The number of voxels for each label and each ROI was then taken as the measure of the effect (Supplementary Table 2 and 5).

In such aggregations, it is incorrect to simply consider all voxels found in a given ROI for all subjects as independent observations. Because of pseudo-replication, in such erroneous approach, an effect (e.g. phonemic based $R^2$ > semantic based $R^2$) found in one single subject with many voxels but not in the other subjects could yield a false result of statistical significance with a grossly deflated p-value. More conservatively, one could first average the data for each subject (e.g. estimating the average "phonemic based $R^2$" - "semantic based $R^2$" for each subject) and perform statistical inference on those subject-averaged values. Although this approach is clearly more correct and conservative, it also gives the same weight to a subject with a small number of significant voxels in the ROI than to a subject that has a higher number of significant voxels in the same ROI. A more principled solution is therefore to explicitly model the statistical dependence of the raw voxelwise $R^2$ or counts in each subject. For this purpose, we used statistical mixed effect models with the subject as the random factor. In mixed effect models, the penalty for the random effect in the overall likelihood that is being maximized for a subject with a smaller number of significant voxels is smaller than for a subject with more significant voxels. In this manner, a likelihood based weighting of the subject data is performed.

In this paper, we used this mixed-effect statistical modeling when reporting the effect sizes and statistical analyses in the text and figures in the main section of the paper. We also repeated all statistical analyses by first performing averages for each subject and then applying classical statistical inference tests on those data. The per-subject analyses and statistics are found in supplementary tables. The statistical results, effect sizes and conclusions obtained using these two approaches were practically identical.

For the statistical analyses presented in the main text, linear mixed-effects statistical models with subjects as random effect (R: lmer) were used to compare the raw $R^2$ for different phonemic and semantic based VMs across in PAC, STG, STS, LTC, Broca's area, and the entire cortex and to test for lateralization effects. Generalized mixed-effects statistical models (R:glmer) with binomial distributions were used to compare the fraction of best explained voxels by the semantic feature based VMs relative to the phonemic feature space VMs across ROIs. Multinomial mixed-effects statistical models (R: mblogit) were used to estimate the fraction of best explained voxels by the single phoneme vs diphone or triphone feature based VM while taking into account the variability across subjects (the random effect). Custom R code was written to calculate the expected likelihood of the null model corresponding to a fixed effect of equal probability ($\frac{1}{3}$) and a random effect yielding a saturated statistical model that predicts the actual empirical probability found for each subject. Likelihood ratio tests were then performed to estimate whether the likelihood of the mixed-effects multinomial statistical model with fitted fixed effects (i.e. non zero intercepts) was greater than the likelihood of the null model. The standard errors in the mixed-effects model were estimated by bootstrapping over subjects. The bootstrap was performed using the emmeans R package for glmer and with custom code for mblogit.

We used Cohen's $d'$ to report the effect sizes of the difference of the mean prediction performance between phonemic and semantic based VM model, as well as the mean prediction performance among single phoneme, diphone and triphone features. In order to obtain the dprime of the phonemic and semantic based VM model, we first used a mixed-effect statistical model with subject as random factor and models as fixed factor. The estimated standard deviation of the random effect of this mixed-effect model captures the variance in mean prediction performance found across subjects, $\sigma^2_{SUB}$; the coefficients of the fixed effects of this model captures the difference in the mean prediction performance between VM models based on different features. For example $\mu_S - \mu_P$ where $\mu_S$ is the mean prediction performance for semantic based VM and $\mu_P$ is the mean prediction performance for the phonemic based VM, and the standard error of these same coefficients gives the standard error in this same mean prediction difference, $\sigma_{SE(S-P)}$, the effect size ($d'$) was then estimated by dividing this difference in average performances by the standard deviation, both estimated in the mixed effect model across subjects:

$$d' = \frac{(\mu_S - \mu_P)}{\sigma_{SUB}} \quad (1)$$

The standard error of the $d'$ was calculated from the standard error for the difference in mean predictions:

$$SE(d') = \frac{\sigma_{SE(S-P)}}{\sigma_{SUB}} \quad (2)$$

In order to obtain the average dprime for diphones relative to single phonemes and diphones VM predictions ($d'_{phn}$), we first calculated the dprime of the diphone versus triphone ($d'_{DiTri}$) and the dprime of the single phoneme versus single phoneme ($d'_{SinDi}$). Then, we simply averaged these two dprimes after correcting for the sign inversion.

$$d'_{phn} = \frac{(d'_{DiTri} - d'_{SinDi})}{2} \quad (3)$$

For the standard deviation of the dprime for diphone VM predictions ($SE(d'_{phn})$), we first computed the sum of the square of the standard error of the dprime of the diphone versus triphone ($SE(d'_{DiTri})$) and diphone versus single phoneme ($SE(d'_{SinDi})$), and then we calculated the half of the square root of this sum.

$$SE(d'_{phn}) = \frac{\sqrt{SE(d'_{DiTri})^2 + SE(d'_{SinDi})^2}}{2} \quad (4)$$

In addition, we also reported a d' for each subject in Supplementary Table 1 and 4. It was calculated as the difference between the means of the average prediction performance between the phonemic and semantic based VM model for each subject divided by the pooled standard deviation. The mean and SE of those per-subject dprimes are summarized in the table as well.

To quantify the difference in the number of voxels that are best explained by phonemic or semantic based VMs, the log of the odds (or logit) that a voxel is labeled as "semantic" (vs "phonemic") was used to calculate the the effect size:

$$l = log\left(\frac{p_S}{1 - p_S}\right) \quad (5)$$

Similarly to compare the number of voxels that are best explained by single phonemes, diphones or triphones, the probability that a voxel is labeled as any of the phonemic label was estimated using a reference in the multinomial statistical model. Here we took the

diphone as the reference and report negative logits, for example for the probability of single phonemes vs diphones:

$$l = -log\left(\frac{p_S}{p_D}\right) \quad (6)$$

The probability values are obtained from mixed effect binomial (semantic versus phonemic) or multinomial (single phonemes vs diphone vs triphone) statistical models in the main text and, just as for $d'$, this calculation is repeated with the "equal weight" per subject and shown in the supplementary tables for Supplementary Table 2 and 5.

## Simulation: feasibility and limitations

In this study we are proposing to use the VM approach to predict BOLD activity based on the segmentation of the speech stream occuring at the level of small phonemic units: single phonemes, diphones and triphones. The temporal resolution of the BOLD signal is known to be particularly slow due to the sluggish hemodynamic response function whose power is concentrated below 0.2 Hz[54]. Here, we investigate the processing of phonemic units which have rates of up to 20 Hz in spoken speech. Stated in this manner, the time scales of phonemes and the BOLD response are mismatched by a factor of 100. Thus, can we prove that our approach could in fact measure correlations between specific phoneme identity (or their combination) and BOLD activity?

As an initial step, one can gain some insight on the feasibility of the approach by examining the signals generated at TR rates for single phonemes or combinations as shown in Fig. 1. Indeed, although the phoneme counts in speech are high, in a given TR there will be a small number of any particular phoneme. One finds mostly counts of zero, one, two or three (and consequently, even smaller numbers of particular diphones and triphones). Moreover, this count changes from TR to TR yielding a clear signal. Thus, as long as voxels are sufficiently sensitive, in the sense of responding more strongly to a small number of phonemes, one should be able to detect this signal in the BOLD response given sufficiently long recording times. To quantify what is "sufficiently sensitive", we thus performed a series of simulations with realistic signal to noise ratio (SNRs) and data size (corresponding to our 2 hour recording time) to evaluate the recovered VM coefficients and compare them to their actual values used to generate the simulated data. The simulation results are based on the SNR values of average responsive voxels (SNR = 1) for subjects with overall high SNR across all voxels (see Supplementary Fig 16). Thus, we are simulating the average case scenario in order to assess the limits of what we would be able to detect inn terms of phonemic feature sensitivity given our data size and for the most responsive voxels.

First, we examined whether our approach could distinguish between voxels sensitive to single phonemes versus those sensitive for diphones, or combinations of single phonemes and diphones. The single phoneme-only simulated voxel was by construction sensitive to three randomly chosen distinct phonemes with equal weights. Similarly, the diphone-only simulated voxel was sensitive to three randomly chosen diphones. The simulated voxel that was sensitive to the combination of single phonemes and diphones had a BOLD signal that was simply the sum of the BOLD signal for the single phoneme and the diphone model voxel. To generate a realistic BOLD signal, the signals obtained from the convolution were low-passed filtered below 0.1 Hz and Gaussian white-noise was added to fit the SNR of one of a highly responsive voxel. The voxel SNR was estimated by calculating the coherence between two repeats of the same speech stream in the validation data set[55]. We repeated this procedure 30 times by resampling among single phonemes and diphones to generate a distribution of simulated results. As shown in Supplementary Fig 3A, for the stimulus duration of 3737 TRs (around 2 hours), we were able to recover

the VM coefficients and expected values for $R^2$. Thus, for average sensitive voxels, our procedure could distinguish BOLD responses reflecting single-phonemes-only sensitivity, diphone-only sensitivity and responses that resulted from sensitivity to the mixtures of particular single phoneme and diphone occurrences.

Next, we examined the effect of voxel sensitivity on our ability to determine whether representations based on phoneme identities would yield better results than those based on just the phoneme count. With the decrease of the sensitivity of putative voxels (in the sense that it responds to a growing number of single phonemes or diphones), we may fail to differentiate between VMs based on phonemic identity versus phonemic rate. As shown in Supplementary Fig 3B, as a voxel's sensitivity decreases (i.e. as it becomes tuned to a larger number of phonemic identities), the predictive power of the VM based on phoneme count increases and approaches that of the VMs based on phoneme identity. The roughly equal prediction performance between the phoneme count and identity based VMs was obtained around 10 single phonemes for the single phoneme based VM and 100 diphones for diphone based VM. Note that the observed decrease of predicted performance of the identity based VMs (green and cyan lines) for low sensitivity voxels is a result of overfitting. Also, the suboptimal performance of the diphone based VM for very sensitive putative voxels (i.e. sensitive for < 10 diphones) is a result of the sparsity of that signal. Particular diphone combinations can occur with low frequency in 2 hours yielding poor estimations of the VM coefficients.

Finally, we investigated the number of phonemic combinations that could be recovered from our dataset. For this purpose, we generated fake data for three model voxels: one voxel was sensitive to the phoneme count, one voxel sensitive to phonemic combinations of increasing order (i.e. single phoneme, diphone, triphone, tetraphone, pentaphone, and hexaphone) and one voxel to the mixture of both. As in our first stimulation, the BOLD signal for the phonemic combination was obtained by randomly assigning weights of three features (i.e. three particular single phonemes, or diphones, etc). Here, taking into account the increasing number of features in each VM, the sampling was repeated 30 times for single phoneme simulation, 50 times for diphones, and 150 times for triphone, tetraphone, pentaphone and hexaphone based VMs to obtain the standard deviation of the prediction performance. As shown in Supplementary Fig 4, we were able to recover the VM coefficients and expected values for $R^2$ of single, diphone and triphone based VMs but unable to recover that of tetraphone, pentaphone and hexaphone based VMs. We also more directly assessed the effect of the data size by comparing the results we obtained for the diphone vs triphone based VM models for a single subject for which we had 5 (5 h) scanning sessions of Moth Radio Hour stimulus. Although the additional predictive power of the triphone based VM relative to the diphone based VM increased slightly, the absolute magnitude of the relative predictive power of the two models, including the dominance of the diphone representation, remained (Supplementary Fig. 15).

Therefore based on these three simulations and on the analysis of one single subject for which we had additional data, with around five hours of narrated speech stimulus (Supplementary Fig. 15), our approach should allow us to identify voxels that show sensitivity for sets of phonemic units and differentiate brain regions that are sensitive to the identity of single phonemes, diphones or triphones as long as they exhibit sufficient sensitivity (approximately for fewer than 10 particular single phonemes and 100 particular diphones). Increasing the duration of the data collection might allow one to assess sensitivity for higher order combinations of phonemes including entire words, or to detect voxels that are sensitive to phoneme identity with lower sensitivity, or to recover voxels sensitive to diphones or triphones with very high sensitivity.

## Reporting summary

Further information on research design is available in the Nature Portfolio Reporting Summary linked to this article.

## Data availability

Raw fMRI data has been made available on https://gin.g-node.org/gallantlab/story_listening. All data other than anatomical brain images has been be shared, as there is concern that anatomical images could violate subject privacy. However, we have also provided matrices that map from volumetric data to cortical flatmaps for visualization purposes. Data generated in this study for each figure are provided in the source data in this repository as well. Source data are provided with this paper.

## Code availability

Custom code used to reproduce the results is available at https://github.com/theunissenlab/phoneme_segmentation.

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

## Acknowledgements

We thank T. Dupré la Tour, S. Popham and A. Nunez-Elizalde for helpful suggestions on data analysis. This work was supported by a Weil Foundation grant to F.E.T., a Dingwall Foundation grant in Neurolinguistics to XG and NSF grant 1912373 to J.L.G.

## Author contributions

X.L.G. and F.E.T. conceived and designed the experiments, analyzed the data and wrote the paper. F.D. and A.H. collected the data. J.L.G. and K.J. contributed the materials and analysis tools.

## Competing interests

The authors declare no competing interests.
