## [Peer Review File · Nature Communications]

Phonemic segmentation of narrative speech in human cerebral cortexReviewer #1 (Remarks to the Author):

Gong et al. presented an fMRI study on how the brain represents phonemes in speech. The main conclusion is that the brain mainly represents diphones, i.e., pairs of phonemes, instead of single phonemes, and the relation between such a diphone representation and word/semantic representations is also discussed. In my opinion, the study presents highly interesting results and the analyses are technically well done. The study more thoroughly considers confounding factors than many other studies, but I still recommend to analyze a few other confounding factors.

1. The psycholinguistic/neurolinguistic literature frequently discusses concepts such as phonemic surprisal, phoneme frequency etc. If the neural response is influenced by, e.g., the occurrence frequency of diphones, it can be better explained by a diphone model compared with a single-phoneme model. Similarly, if phonemic surprisal contains contextual information and can result in something that is highly related to diphones. I wonder whether neural activity truly encodes the content of diphones or some statistical properties related to diphones or the phonemic sequence. Adding some statistical features into regressors or some discussions are needed.

2. A brief discussion is needed about whether the current result may reflect a mechanism specific to speech processing in noise. When listening in (scanner) noise, speech comprehension can be facilitated by the context, which is a possible reason why diphones are more reliably represented than single phonemes?

3. It's true that the triphone model works less well than the diphone model, but a possibility is that the model has too many parameters and therefore is difficult to fit. Maybe if the dataset is even bigger, the triphone model will outperform the diphone model? Furthermore, if I understood correctly, the triphone model simply includes more features than the diphone model. Therefore, if the model is really smart, it can simply ignore, e.g., the last phone in the triphone combination and perform as well as the diphone model.

4. I didn't understand the control on lexical retrieval. More specifically, I didn't figure out how the contribution of the each diphone category is estimated. The 3 categories occur in nonoverlapping time intervals. Are they separately used as the regressors? If so, I wonder if the each of the 3 regressors can generalize to the other 2 categories.

5. It's possible that the responses to vowels can be easily discriminated from the responses to consonants, and it's much harder to discriminate the responses to two vowels or two consonants. I wonder if the phoneme model significantly outperforms a simpler model that only discriminates vowels and consonants. If the phoneme model outperforms the V/C model, I wonder if vowels or consonants contribute more to phoneme decoding.

Minor issues:

1. I wonder how the diphones and triphones are encoded. one-hot coding?
2. I think "phone count" is clearer than "phoneme rate features" and can be consistently used in the manuscript.
3. Simply calling VM "model" can probably facilitate reading.
4. I don't quite follow the argument that "The over-representation for diphones over single phones documented here could be the result of their higher correlation with the speech envelope." The study removed neural responses related to spectrotemporal features, and the envelope can be easily explained by these features.
5. R^2 is reported for the phoneme models (Fig. 3) but not for the baseline models. Please also report the R^2 for baseline models.

Reviewer #2 (Remarks to the Author):

The article by Gong et al. investigates speech processing in human cortex using a data driven regression approach. Starting from the hypothesis that speech processing progresses hierarchically from acoustic to phonemic to word level corresponding feature spaces are constructed to probe the sensitivity of BOLD responses in fMRI-voxels to those features in nested regression models. The authors report feature preferences for various brain areas and two anatomical gradients along which hierarchical processing (phonetic to semantic) may develop. They also report functional overlap in the areas that are hard to demonstrate with traditional approaches. The study is methodologically demanding and provides very interesting insights partly confirming, and partly exceeding previous studies. I consider it an important contribution to the field that has implications for theories of speech processing in human cortex. However, I have some questions/remarks that should be addressed.

My main points are mostly methodological:

I may have missed something but it seems that only one test block was used to derive the results. However, the construction of the experiment would allow for a full eleven-fold cross validation across blocks. This would greatly improve the validity of the results of a very complex model fitting procedure. I'm aware that the test stimulus was repeated to calculate an adjusted prediction for the sensitivity maps but the statistics was calculated on the raw R^2 values and would benefit from an improved cross validation.

The permutation stats approach appears not sufficiently specific for the conclusions in the paper. The reference distribution of R^2 values is obtained by shuffling blocks of the predictions of the validation story. This reference distribution can be used to test whether the predicted/measured response R^2 is obtained by chance but does not specifically test whether the speech features make the model explain variance. Thus I suggest to rather shuffle features for model training and derive the R^2 reference distribution from the predictions with models trained on the permuted features. This is more specific for the conclusions drawn about sensitivity to speech features. The current permutations approach destroys any correlation, be it due to successful prediction of speech features or anything other information the complex model was able to pick up.

The finding that areas with high phonemic sensitivity are particular sensitive to diphones as compared to phonemes or triphones is very interesting and I very much appreciated the control for sensitivity for short words. These and other findings are based on the variance partitioning approach. However, I have some problems with the assumption that the variance partitioning simply reflects partialized variance estimates in this study. This may only hold for standard OLS solutions with uncorrelated regressors. It is unlikely that the regressors are uncorrelated in this study. Correlations may exist even across feature blocks. The problem of multicollinearity is that model coefficients may change when correlated regressors are removed but the R^2 may change only little. this means that the sensitivity of the model to certain (e.g. phonetic) features may change but the R^2 remains similar. L2 regularization in ridge regression does not prevent this as does not separate the effects of correlated regressors but smears the weights across them. Thus if regressors are correlated across feature blocks fitting the reduced model may compensate the loss of some correlated regressors by upscaling the weights of the remaining regressors in the reduced model. Importantly, R^2 is only proxy measure for feature sensitivity of the model. I think such problems were already at least partly addressed in de Heer et al 2017. Moreover, the fact that the limited amount of data cannot span the full feature space and separate regularization factors across blocks fitted separately for the full and reduced models seem to further complicate the interpretation of R^2 changes as the proportion of variance attributed to a feature block. The banded regression is a smart approach to allow for different regularizations for the feature blocks to improve the model fit but it may also produce additional deviations from the variance partitioning assumption and have an unpredictable influence on the model coefficients when regressors are removed. A critical discussion of these limitations the approach imposes on the conclusions or, even

better, an assessment of the severity of such potential effects would be helpful.

In the same line of reasoning I wonder why the main analysis was performed on the residuals of acoustic+phoneme rate model. Why were these features not simply included in the models of the main analysis?

I appreciate the simulation as it provides interesting results. However, I think its sensitivity estimates are too optimistic for the encoding models trained on real data and corrupted by non-stationarity of brains over days, realignment limitations, etc. This is should be critically discussed.

The language is sometimes too strong and inadequate. For example on p.4 the authors write "In order to localize the cortical regions uniquely selective to phonemic processing ..." I do not think that the methods used here, though powerful and impressive, allow such statements. First, the study only uses speech sounds. Sounds from other generators may drive these areas as well. Thus functional specialization for speech of brain regions cannot be concluded from the experiments. I suggest to rephrase such statements throughout the manuscript. Moreover, selectivity is a strong concept suggesting that voxels discriminate stimulus features. However, the approach used here can only demonstrate sensitivity (see my comment on regressor correlation above.) I suggest to replace selectivity with sensitivity throughout the manuscript. The term sensitivity leaves the question of discrimination open.

I wonder if the functional segregation between semantic and phonemic processing is really as categorical as figures 6A and B suggest. The choice of the colors may mislead the reader as diagonal is gray along all levels of sensitivity. Figure 6D suggests preference rather than categorical difference in most areas. It may be helpful to use a more continuous color scale (e.g. red, yellow, green) that lets the reader appreciate the degree of multiplexing. One advantage of the method used here is that it can quantify functional overlap.

If no other restrictions prohibit it the authors should add a note on how the interested reader can obtain code and data to reproduce the results.

Minor comments:

p.19 "Our simulations (Methods), however, show that the capability of detecting fast neural activities underlying speech processing using fMRI is not necessarily constrained by this low temporal resolution as long as the fast speech events being studied are temporally sparse." I do not think they need to be temporally sparse. They only need to be sufficiently variable at the right temporal scale. At least for low dimensional models. Sparseness may help to with the problem multicollinearity in big models.

p. 19 Was there any control to ensure the participants listened to the stories?

p.20 What is the "overall" template? Over all runs but within participant or over all subjects and runs?

p.21 what is the "overall" template? Over all runs but within participant or over all subjects and runs?

p.23 Typo R2measuring

Reviewer #3 (Remarks to the Author):

In this manuscript, Gong and colleagues employ an encoding method to study voxels' sensitivity to features composing auditory stimuli. In this sort of research strategies, subjects listen to continuous speech over the course of the session, typically long

periods. The set of features characterizing the stimuli can then be used in a multiple regression model using Ridge regularization and cross-validation schemes. A good model should allow generalizing the model to new data. Gong used this type of models to predict the fMRI signal using different sets of features in 11 subjects, from acoustic spectral features to phonetic features (phonemes, diphones and triphones) and semantic features, as well as, their hierarchical relations using partial variance explained by embedded feature sets.

The study is very well presented. Very well written and make use of high quality visuals (figures) to communicate their methods and results. The analysis strategy is well known, as it is almost the same as in Heer et al (2017).

Despite the several positive remarks I have already mentioned, I find the study lacks novelty. It is true that it is interesting to study the continuum that speech comprehension encompasses (from acoustic perception to semantic access), and that brain differences along this continuum may be related to how humans make sense of speech. However, other linguistic characteristics overlap with phonemic size, for example phonemic frequency, lexicality, semantic neighbors size, etc.

This study is very well thought, executed and written, but given the several other studies using the exact same stimuli and analysis, the current study would benefit from some novel aspect that could validate the author's conclusions.

For example: If subjects would also listen to a similar story in a language they don't understand, we should expect very different results in the access of semantic knowledge.

I also have a couple of remarks that could be better explained in the manuscript:

1- The cross-validation method involves cutting epochs of 20 seconds (10 TRs) and permuting them to assess chance level in fMRI signal prediction based on the training model parameters. In Heer 2017, 40 seconds were used for the same goal. WHY using a smaller window size for the permutations? And why now permute the labels instead (i.e., the story ID). I fear that this strategy to permute windows of data may have consequences to the low frequencies that characterize the BOLD signal (as the authors elude in the method's section), by introducing abrupt signal changes that the original (un-permuted) signal doesn't have.

2- WHY exactly is the R2 used for model validation instead of the r ? If I understand correctly, the model is employed to predict the fMRI response, and correlations verify whether the prediction and real signal are similar. If so, one expects positive correlations (r), and R2 hides this. I understand that R2 speaks for variance explained and it should still be expected that the permuted data should have less variance explained. So, my remark (question) is more on the conceptual aspect of how to employ this encoding models. Positive correlations are meaningful and negative correlations should not be meaningful. If this is correct, by hiding the sign of the correlation (say in the brain maps) may create clusters of voxels with opposite r signs.

References:

Heer, Wendy A. de, Alexander G. Huth, Thomas L. Griffiths, Jack L. Gallant, and Frédéric E. Theunissen. 2017. "The Hierarchical Cortical Organization of Human Speech Processing." *Journal of Neuroscience* 37 (27): 6539–57.
<https://doi.org/10.1523/JNEUROSCI.3267-16.2017>.

Response to Reviewers:

We thank the reviewers for their thorough reviews and their candid assessment of our contribution. We believe that we have addressed all of their comments. The reviewers' comments are italicized and bolded and our responses are in normal text.

Reviewer #1

Reviewer 1 summarized nicely the results from our study in the introductory paragraphs of the review:

“In my opinion, the study presents highly interesting results and the analyses are technically well done.”

Thank you.

In the next paragraph some weaknesses in terms of confounding factors are stated:

“I wonder whether neural activity truly encodes the content of diphones or some statistical properties related to diphones or the phonemic sequence. Adding some statistical features into regressors or some discussions are needed.”

This is a good point. The expectations of phonemic sequences are also captured by diphones models and could be the driving speech feature leading to the increase in predictive power that we observed with diphones identity models, in particular since each of our TRs are composed of multiple diphones and predictions are obtained for 4 TR. To disentangle the fraction of the predictive power that might be solely due to the diphone identity versus the fraction due to expected sequences of diphones, we fitted additional models that used only the statistical expectations of diphone sequences. The novel analysis is found in Supplementary Figure 9 and Supplementary Table 6 and is referred to in the text in the section now titled: “Diphone segmentation for identification, expectations or lexical retrieval?”.

The supplemental method section states:

A large body of work has shown the importance of predictions in speech processing (Leonard et al, 2015, Heilbron et al, 2022, Di Liberto et al, 2019). In order to further explore if the cortical BOLD activities truly encode the content of diphones or merely the statistical properties of diphones, we built one diphone statistics model consisting of 8 phonological statistical features extracted from Irvine Phonotactic Online Dictionary (IPhOD; Vaden, 2009). These features describe the phonotactic probability of diphones. The table summarizes the name and content of each feature.

Feature Name	Feature Definition
unsDPAV	Unstressed diphone probability average; vowel-stress ignored
unsFDPAV	unsDPAV weighted with SUBTLEXus word frequency
unsLDPAV	unsDPAV weighted with log SUBTLEXus word frequency
unsCDPAV	unsDPAV weighted with SUBTLEXus context count
strDPAV	Stressed diphone probability average; distinct stressed-vowels
strFDPAV	strDPAV weighted with SUBTLEXus word frequency
strLDPAV	strDPAV weighted with log SUBTLEXus word frequency
strCDPAV	strDPAV weighted with SUBTLEXus context count

Phonotactic probabilities refer to the concurrence likelihood of the sequence of sounds that are present in a given word (Vaden 2009; Vitevitch, 2004). Diphone probability average refers to the average likelihood of each diphone occurring in each position of a word. In these measures, the syllable stress placement of vowels can also be considered. For stressed calculations, identical vowel sounds are considered to be distinct phonemes depending on primary, secondary, or no-stress placement. In unstressed calculations, vowel sounds are considered to be single phoneme categories.

The box plot (A) shows the average prediction performance of all the significant cerebral cortex voxels of 11 subjects (each dot represents one subject) from the diphone statistics model (orange), the diphone identity model (blue), and both models (green). It shows that the performance of the diphone statistics model is significantly lower than that of the diphone identity model (blue) ($t(1) = -30.19$, $p < 2.2 \times 10^{-16}$). The flatmap (B) shows the anatomical data for one example subject (S5). Blue voxels are better predicted by the diphone (content) model, and orange voxels are better predicted by the diphone statistics model. White voxels are equally well predicted by both models. This analysis shows that although diphone statistical properties can explain some of the variance of BOLD responses that could be captured by the diphone model, the actual diphone identities captured in the diphone model and not in the phonotactic probabilities yield significant additional explanatory power.

Reference:

Heilbron, M., Armeni, K., Schoffelen, J. M., Hagoort, P., & De Lange, F. P. (2022). A hierarchy of linguistic predictions during natural language comprehension. *Proceedings of the National Academy of Sciences*, 119(32), e2201968119.

Leonard, M. K., Bouchard, K. E., Tang, C., & Chang, E. F. (2015). Dynamic encoding of speech sequence probability in human temporal cortex. *Journal of Neuroscience*, 35(18), 7203-7214.

Di Liberto, G. M., Wong, D., Melnik, G. A., & de Cheveigné, A. (2019). Low-frequency cortical responses to natural speech reflect probabilistic phonotactics. *Neuroimage*, 196, 237-247.

Vaden, K.I., Halpin, H.R., Hickok, G.S. (2009). Irvine Phonotactic Online Dictionary, Version 2.0. [Data file]. Available from <http://www.iphod.com>

Vitevitch, M.S. & Luce, P.A. (2004). A web-based interface to calculate phonotactic probability for words and nonwords in English. *Behavior Research Methods, Instruments, and Computers*, 36, 481-487.

https://corpustools.readthedocs.io/en/latest/phonotactic_probability.html

Brysbaert, M., & New, B. (2009). Moving beyond Kučera and Francis: A critical evaluation of current word frequency norms and the introduction of a new and improved word frequency measure for American English. *Behavior research methods*, 41(4), 977-990.

“A brief discussion is needed about whether the current result may reflect a mechanism specific to speech processing in noise. When listening in (scanner) noise, speech comprehension can be facilitated by the context, which is a possible reason why diphones are more reliably represented than single phonemes?”

We agree with the reviewer that we are testing speech perception in noise and that this caveat should also be addressed. We now added to the paragraph in the discussion where we list the caveats, the following clause:

"Finally, the dominance of diphones may also reflect the engagement of predictive speech processing mechanisms which become more important in noisy situations (Trecca et al, 2019). In particular, under the noisy fMRI scanning environment, diphones could be more reliably represented and resistant to noise than single phonemes. On the other hand, the predictive power of VM that were solely based on the statistical properties of diphone sequences and ignored diphone identities was significantly smaller (see supplemental Figure 9).

Reference:

Trecca, F., Tylén, K., Fusaroli, R., Johansson, C., & Christiansen, M. H. (2019). Top-down information is more important in noisy situations: Exploring the role of pragmatic, semantic, and syntactic information in language processing.

“ It’s true that the triphone model works less well than the diphone model, but a possibility is that the model has too many parameters and therefore is difficult to fit. Maybe if the dataset is even bigger, the triphone model will outperform the diphone model? Furthermore, if I understood correctly, the triphone model simply includes more features than the diphone model. Therefore, if the model is really smart, it can simply ignore, e.g., the last phone in the triphone combination and perform as well as the diphone model.”

Our nested models analysis was designed to address this possibility: the third order phonemic model (which includes single phones, diphones and triphones) is the triphone model that we believe the reviewer has in mind. It includes all the diphones and all the additional phonemic structure that one can extract from the particular triphone identity beyond what one can obtain from diphones identity. The predictive power of the third order model is indeed always greater than that of the second-order model. In an earlier version of the manuscript, we also showed the flat maps obtained from these nested models first order phonemic model (single phoneme features), second order phonemic model (single phoneme + diphone features), and third order phonemic model (single phoneme + diphone + triphone features). We realize now that illustrating our methodology by also reporting these results is useful. We have therefore added a supplement figure 8 that shows the prediction performance from all nested models of all subjects. It clearly shows that the third order phonemic model indeed outperforms the second order phonemic model.

Nonetheless, the concern regarding the effect of the data size remains valid. The small increase in predictive power of the third order model might be a consequence of its large number of parameters relative to the size of the data. In order to explore this effect, we also replicated our analyses on one subject from a larger dataset using 5 sessions of moth stories. This data has 9461TR training data (~5 hours) and the same test data. (<https://openneuro.org/datasets/ds003020/versions/1.0.2>).

The bar plot(A) below shows the average prediction performance (R^2) across significant voxels obtained from second order phonemic model (single phoneme + diphone: cyan), third order phonemic model (single phoneme + diphone + triphone: yellow), diphone only (blue) and triphone only (orange) It reveals that diphone features’ prediction performance is significantly higher than the predictions obtained from triphone features (blue). This effect is observed both for analyses based on 2 sessions and 5 sessions of data collection. There is a small difference in terms of additional prediction obtained from the triphone features after taking into account diphone features when more data is used. Nonetheless, our central result remains: the diphone features play a more important role than the triphone features for predicting the bold response in phonemic cortical regions.

The dominance of the diphone features is clearly seen in the flatmaps (B). The blue voxels are better explained by the diphone features, while the orange voxels are better explained by the

triphone features. White voxels are equally well predicted by diphone and triphone features. The flatmap shows more blue voxels than orange or white voxels irrespective of whether 2 or 5 sessions or data are used. We have now added this analysis as a supplemental figure 15 and briefly describe the results in the methods section where we addressed the feasibility and limitations of this procedure.

“4. I didn’t understand the control on lexical retrieval. More specifically, I didn’t figure out how the contribution of the each diphone category is estimated. The 3 categories occur in nonoverlapping time intervals. Are they separately used as the regressors? If so, I wonder if the each of the 3 regressors can generalize to the other 2 categories.”

Thank you for this comment. This section was indeed poorly explained. For this analysis, we did not fit a new VM with additional categories of diphones as regressors. Instead, we examined the predictions obtained separately by diphones that were words, beginning of words or others. To obtain these predictions, we just set zero to the diphone vector for diphones belonging to the category that was not being considered. We have revised that section in the methods. It now says:

“In a separate analysis, we examined the contribution of distinct types of diphone features. First, we separated diphone into three categories: diphones that are short words (e.g. “M.AY” as word “my”), diphones that are at the beginning of words (e.g. “AE.N” in word “and”), and all the other diphones. We then estimated the contribution of each diphone category to the prediction by deleting the diphones that did not belong to the category being considered. More precisely, we set the contribution to the diphone feature vector from the out-of-considered-category diphones to zero. With this procedure, we divided the predicted signal obtained from the diphones into three components, one obtained for each mutually exclusive diphone category. “

“5. It’s possible that the responses to vowels can be easily discriminated from the responses to consonants, and it’s much harder to discriminate the responses to two vowels or two consonants. I wonder if the phoneme model significantly outperforms a simpler model that only discriminates vowels and consonants. If the phoneme model outperforms the V/C model, I wonder if vowels or consonants contribute more to phoneme decoding.”

These are very good questions that we are also interested in answering. Although performing a full analysis of phonemic model weights is beyond the scope of this paper (something we are pursuing in a stepwise manner), it is relatively simple to address this particular query by testing the contribution of VM that are based on the category of phonemes in the vowel/consonant set.

In the revised manuscript, we added an in depth analysis of the cortical representation of vowel and consonants (V/C) and combinations of vowel and consonant as diphones. To not overwhelm the results presented in the original paper with this additional analysis, we decided to add it as supplementary material and refer to it in the main text.

In the supplementary material (supplemental figure 10 and supplemental methods), we have added the following:

“In order to begin to explore the distinct roles of vowels and consonants in the cortical representation of phonemes and phonemic combinations, we created feature spaces based on the identity of consonants and vowels of each phoneme.

The single_vc feature is a one-hot coding matrix (dimension: [number of TR, 2]) to encode the presence of a single phoneme and whether it is a consonant or vowel.

The diphone_vc feature (dimension: [number of TR, 6]) encodes the presence of a diphone and its belonging to one of these six consonant/vowel combinations: vowel-vowel (vv),

vowel-consonant (vc), consonant-vowel (cv) and consonant-consonant (cc), vowel-blank (vb), consonant-blank (cb)

We then fitted four nested VMs: single vc model (using single_vc feature), first order vc model (using single_vc + single phoneme features), first order vc + diphone vc model (using single_vc + single phoneme + diphone_vc features) and second order vc model (using single_vc + single phoneme + diphone_vc + diphone features). The variance explained by the single_vc feature is obtained from the single vc model, while the additional variance explained by diphone_vc feature is obtained from subtracting the explainable variance of the first order vc model from the explainable variance of the first order vc + diphone vc model. In addition, the additional variance explained by the single phoneme identities beyond vowel and consonant categories is obtained from subtracting the explainable variance of the single vc model (using single_vc feature) from the first order vc model. Similarly, the variance explained by the diphone identities beyond the combination of vowel and consonants is obtained from subtracting the explainable variance of the first order vc + diphone vc model from the second order vc model.

The box plot (A) shows the average additive prediction performance across the significant cerebral cortical voxels (each dot represents one subject) obtained for single phoneme features, single_vc features, diphone features and diphone_vc features. The flatmap (B) shows the additive prediction performance of each feature from one example subject (S5). This result illustrates that the phoneme identity models (single phoneme and diphone features in the paper) significantly outperform the vowel and consonants features.

In order to compare the relative contribution of vowels and consonants towards the phoneme encoding, we examined the modeling weights (coefficients) of single_vc (consonant or vowels) and diphone_vc (vv, vc, cv, cc, vb, cb) features. The weights are obtained from the second order vc model. The box plot (C) shows the average (across voxels and subjects) of the absolute value of the weights for each regressor/channel of the single_vc and diphone_vc features. The weights of each voxel have been scaled by the prediction performance of this voxel in order to get rid of the random effects from noisy voxels. The box plots show that there is no significant difference in the contribution consonants and vowels towards the single phoneme encoding. For diphones, “vc” and “cv” combinations contribute the most to the variance explained.. “

B

Models

C

Minor Comments

1. I wonder how the diphones and triphones are encoded. one-hot coding?

Yes, the first step of feature construction for diphones and triphones is one-hot encoding. We revised the corresponding description in the “Feature Space and Model Construction” subsection of the method section. Afterwards, the matrix was downsampled.

2. I think “phone count” is clearer than “phoneme rate features” and can be consistently used in the manuscript.

We have revised the manuscript to fix this inconsistency. Thank you

3. Simply calling VM “model” can probably facilitate reading.

Using “model” might indeed facilitate the reading flow but we would like to keep the VM notation to be consistent with our prior publications.

4. I don't quite follow the argument that “The over-representation for diphones over single phones documented here could be the result of their higher correlation with the speech envelope.” The study removed neural responses related to spectrotemporal features, and the envelope can be easily explained by these features.

The reviewer is correct. We removed this sentence.

5. R² is reported for the phoneme models (Fig. 3) but not for the baseline models. Please also report the R² for baseline models.

The R² for the baseline is shown on Figure 2 Baseline flatmap for one example subject and supplementary figure 5 row 1 for all the other subjects.

Reviewer #2

“The study is methodologically demanding and provides very interesting insights partly confirming, and partly exceeding previous studies. I consider it an important contribution to the field that has implications for theories of speech processing in human cortex.”

Thank you for the positive comment.

“I may have missed something but it seems that only one test block was used to derive the results. However, the construction of the experiment would allow for a full eleven-fold

cross validation across blocks. This would greatly improve the validity of the results of a very complex model fitting procedure. I'm aware that the test stimulus was repeated to calculate an adjusted prediction for the sensitivity maps but the statistics was calculated on the raw R² values and would benefit from an improved cross validation."

We agree with the reviewer that a full CV could provide more accurate estimates of R² as long as it is performed correctly. We have adopted the particular approach of having only a single cross-validation testing stimulus (one story) for the following reason. We use the full CV with the fitting set of stories to find the best values for the hyperparameters of the block-ridge regression: for each potential value of the hyperparameter, we fit the model weights (or. coefficients) on 9 of the 10 folds of the fitting set and measure prediction on the left-out fold. We repeat this for the 10 folds (and all values of hyperparameters). Using this procedure we choose the hyperparameter values that gave the best CV predictions. We then perform one more cross-validation on the testing story, a untouched stimulus-response dataset (not used to fit either the model coefficients or the hyperparameters). It is true that one could envision performing a more complex CV-fold procedure that divides all the data into three sets, one for fitting the model-coefficients, one for choosing the hyperparameters based on the predictive power in that second partition and one to obtain the final unbiased R² on a third partition. However, one should note that the multiple obtained values of prediction for the hyperparameters and final R² estimates are not independent in such a procedure. Adjusted statistical procedures need to be used when generating confidence intervals from such data. Although we are aware of approaches that deal with the non-independence for a simple CV (with fitting folds and one testing folds), we are not aware of a good statistical analysis that shows how to obtain reliability estimates using the double testing sets (one of hyperparameters and one for the unbiased R²) as it is required here. Thus, we opted for the most conservative approach of reserving a relatively long testing set for the sole purpose of estimating the unbiased R². The statistical significance of that estimate could then be assessed by permuting the data on this testing data set. Again it is a conservative approach that might lower the confidence on our estimated R² (which we can only provide indirectly using statistical significance instead of confidence intervals) but for which we can be confident that we are not overestimating either the absolute R² or its confidence intervals.

We performed the following analysis to illustrate the potential bias in a regular CV-fold approach. The bar plot empirically shows the average prediction performance of all the significant voxels from one example subject (S5) of the toSemAll model calculated from tenfold cross validation by using the all 11 stories (blue bar), and the average prediction performance by tenfold cross validation on 10 stories and a separate test data (orange bar: our paper method). The average performance of the 10CV+test model is significantly lower than the 11CV model. The flatmap shows the prediction performance for each voxel from the 11CV model and 10CV+test model. Blue voxels are better predicted by the 11CV model, orange voxels are better predicted by the 10CV+test model and the white voxels are equally predicted by both models. These results empirically support the previous explanation that the method we used in the paper is more conservative and has higher validity in generalization.

We have also slightly revised our methods section describing our CV procedure so that it more explicitly describes this two stage process.

“In general, our modeling method has two phases. First is the training phase, second is the testing phase. First, we separated the data into training data (3737 TR) and test data (291TR). For estimating the models (meaning selecting hyperparameters, and calculating the weights), we only use the training data (3737 TR). During the training phase, we used tenfold cross validation (repeated for 10 times) to estimate the weights and hyperparameters in the training data only. It means that every time, we separate 3737 TR into training (2937 TR) and validation data (800 TR). We repeated this for 10 times. After we obtain the optimal weights and

hyperparameters using this tenfold cross validation method, we enter the testing phase. We test the weights and hyperparameters in a totally separate and unused test data (291TR, our test story). In this way, we are essentially implementing a much stricter generalization/statistical testing. We estimate the statistical significance on this totally new test dataset.”

“The permutation stats approach appears not sufficiently specific for the conclusions in the paper. The reference distribution of R^2 values is obtained by shuffling blocks of the predictions of the validation story. This reference distribution can be used to test whether the predicted/measured response R^2 is obtained by chance but does not specifically test whether the speech features make the model explain variance. Thus I suggest to rather shuffle features for model training and derive the R^2 reference distribution from the predictions with models trained on the permuted features. This is more specific for the conclusions drawn about sensitivity to speech features. The current permutations approach destroys any correlation, be it due to successful prediction of speech features or anything other information the complex model was able to pick up.”

We agree with the reviewer that one could shuffle features prior to model training to obtain the null distribution of the R^2 values as it is possible that the stimulus shuffling fails to break long-term correlations that remain explained by the model.

Therefore, we reran all the analysis and statistics using the permutation method proposed by the reviewer. We revised the permutation analysis part in the method section:

“The statistical significance of the cross-validated coefficient of determination R^2 was estimated using a permutation analysis. First, We obtained a set of regularization hyperparameters from fitting the phoneme-semantic VM (using single phoneme, diphone, triphone and semantic features) for each subject. Then, a permuted R^2 for each model of each subject was obtained by refitting the model using this set of hyperparameters and the shuffled regressors of each feature space within the model. This process was repeated 1,000 times to generate the null distribution of cross-validated R^2 for each voxel of each subject. Based on the values of this null distribution, we used variable thresholds yielding a significance level of 1% corrected using the False Discovery Rate (FDR) procedure (Benjamini, 1995) to determine statistical significance.”

The results obtained with this new permutation test (permutation of features) are very similar to those that we obtained in the original permutation test (permutation of predictions) and the conclusions identical. The permutation of features test resulted in slightly more conservative results resulting in fewer significant voxels for all models. The reviewer will also note (by reading the versions showing the comparison between old and new manuscript) that by restricting our analyses to this slightly smaller number of significant voxels that all our effect sizes became larger, presumably because we are now focusing on voxels with larger predictive power. It is clear that the feature permutation is a superior statistical test. It is however much more computing intensive, which is why we had not used it initially. The computing time needed to rerun all of these analyses with this permutation test (and the regeneration of all figures and

recalculation of subsequent statistical analyses) is also the principal reason for the relatively long delay in our response to this review.

“The finding that areas with high phonemic sensitivity are particular sensitive to diphones as compared to phonemes or triphones is very interesting and I very much appreciated the control for sensitivity for short words. These and other findings are based on the variance partitioning approach. However, I have some problems with the assumption that the variance partitioning simply reflects partialized variance estimates in this study. This may only hold for standard OLS solutions with uncorrelated regressors. It is unlikely that the regressors are uncorrelated in this study. Correlations may exist even across feature blocks. The problem of multicollinearity is that model coefficients may change when correlated regressors are removed but the R^2 may change only little. this means that the sensitivity of the model to certain (e.g. phonetic) features may change but the R^2 remains similar. L2 regularization in ridge regression does not prevent this as does not separate the effects of correlated regressors but smears the weights across them. Thus if regressors are correlated across feature blocks fitting the reduced model may compensate the loss of some correlated regressors by upscaling the weights of the remaining regressors in the reduced model. Importantly, R^2 is only proxy measure for feature sensitivity of the model. I think such problems were already at least partly addressed in de Heer et al 2017. Moreover, the fact that the limited amount of data cannot span the full feature space and separate regularization factors across blocks fitted separately for the full and reduced models seem to further complicate the interpretation of R^2 changes as the proportion of variance attributed to a feature block. The banded regression is a smart approach to allow for different regularizations for the feature blocks to improve the model fit but it may also produce additional deviations from the variance partitioning assumption and have an unpredictable influence on the model coefficients when regressors are removed. A critical discussion of these limitations the approach imposes on the conclusions or, even better, an assessment of the severity of such potential effects would be helpful.”

These are valid concerns that we also appreciate as the reviewer knows but that were not well addressed in this manuscript. In the revised version, we first replicated the methodology used in de Heer 2017 and then performed a variance correction. We believe that this is currently the most rigorous approach for implementing variance partitioning with regularized models. We updated the figure 3 and all the analysis concerning phonemic analysis using this new method. Also, we revised the variance partitioning part of the method section with a more detailed explanation of this methodology:

“In order to quantify the variance and localize the cortical regions uniquely explained by individual feature spaces and any combination of feature spaces, we performed a variance partitioning procedure. We first fitted a joint phonemic model with all phonemic features (single phonemes, diphones and triphones). Then, we used the hyperparameters (to the optimal

shrinkage) obtained from this joint phonemic model of each block of features in the banded regression so that the same subspace of features obtained by the ridge is used in all models. We compute the variance of seven possible partitions: the unique variance of each feature and all the possible combinations of these features (single phonemes + diphones, diphones + triphones, single phonemes + triphone, single phonemes + diphones + triphones). Afterwards, we perform a variance correction on R^2 that is used to eliminate biases in the R^2 estimation that could lead to nonsensical results. We believe that this is currently the most rigorous approach for implementing variance partitioning with regularized models. ”

“In the same line of reasoning I wonder why the main analysis was performed on the residuals of acoustic+phoneme rate model. Why were these features not simply included in the models of the main analysis?”

First, we chose to perform the main analysis on the residuals of the acoustic + phoneme rate model, so that we could focus our analysis/results on the phonemic and semantic processing.

Second, we consider that performing the main analysis on the residuals is conceptually comparable to the traditional block design method in speech research using fMRI. One of the purposes of traditional block design is to control the low-level acoustic and phonemic information, which is comparable to the stepwise regression analysis of acoustic+phoneme rate model in the paper. In this way, we actually showed the power of voxelwise modeling that we could regress out the features/factors that we would like to control and focus on the features/factors that we are interested in.

Nonetheless, the reviewer is correct to wonder whether not using the stepwise regression procedure and instead including the acoustic features could have changed our conclusions regarding phonemic processing. Thus, we performed an additional analysis. We created 4 models: `baseline_firstOrder_model` (acoustic + phoneme count + single phoneme features), `baseline_secondOrder_model` (acoustic + phoneme count + single phoneme + diphone features), `baseline_thirdOrder_model` (acoustic + phoneme count + single phoneme + diphone + triphone features) and `baseline_toSemAll_model` (acoustic + phoneme count + single phoneme + diphone + triphone + semantic features). We then used variance partitioning to obtain the performance for single phoneme (`baseline_firstOrder_model` - baseline model), diphone phoneme (`baseline_secondOrder_model` - `baseline_firstOrder_model`), triphone phoneme (`baseline_thirdOrder_model` - `baseline_secondOrder_model`), and semantic features (`baseline_toSemAll_model` - `baseline_thirdOrder_model`), which is the same way as we used in the paper just without the baseline model (acoustic + phoneme rate features) incorporated. The bar plot shows the average prediction performance from one subject (S5) across all significant cerebral cortical voxels for single phoneme, diphone, triphone and semantic features with (blue bars) and without (orange bars) baseline model. Adding the acoustic features slightly increases the prediction performance in all cases but our main effect remains with similar effect size (or if anything slightly larger): the average prediction performance from diphone features is significantly higher than the single phoneme and triphone features.

“I appreciate the simulation as it provides interesting results. However, I think its sensitivity estimates are too optimistic for the encoding models trained on real data and corrupted by non-stationarity of brains over days, realignment limitations, etc. This should be critically discussed.”

We agree with the reviewer that it is important to understand the SNR of the real data in order to properly evaluate the simulation results. We have added the following text to the subsection *Simulation: Feasibility and Limitations* of the Method section and the figure to the supplemental figure 16.

“To quantify what is “sufficiently sensitive”, we thus performed a series of simulations with realistic signal to noise ratio (SNRs) and data size (corresponding to our ~2h recording time) to evaluate the recovered VM coefficients and compare them to their actual values used to generate the simulated data. We calculated the empirical SNR for each voxel of each subject (histogram below: each color represents one subject). The SNR is obtained from computing the coefficient of determination between two repeats of the BOLD responses collected when the subject was listening to the same story. Based on the empirical SNR of our dataset, we chose SNR=1 for the simulation. Thus, we are simulating the typical/median case scenario in order to assess what we would be able to detect in terms of phonemic feature sensitivity given our data size for most voxels”

The plot here shows the two repeats of the BOLD response of a typical/median responsive voxel (SNR=1.08).

As a separate note, we go to great lengths to limit fluctuations due to physiological non-stationarity or to changes across scanning sections. For example: each subject uses a customized head case to eliminate the head motion during and across sessions. For BOLD signal preprocessing, the cardiac and respiratory signals, and instrumental drifts trend were regressed out. Manual check was employed to eliminate any corrupted data from hardware instabilities (e.g. spiking). We therefore obtain relatively constant SNRs for a given voxel. But as mentioned above we completely agree that the SNR value used for the simulations, represented a good case scenario.

“The language is sometimes too strong and inadequate. For example on p.4 the authors write “In order to localize the cortical regions uniquely selective to phonemic processing ...” I do not think that the methods used here, though powerful and impressive, allow such statements. First, the study only uses speech sounds. Sounds from other generators may drive these areas as well. Thus functional specialization for speech of brain regions cannot be concluded from the experiments. I suggest to rephrase such statements throughout the manuscript. Moreover, selectivity is a strong concept suggesting that voxels discriminate stimulus features. However, the approach used here can only demonstrate sensitivity (see my comment on regressor correlation above.) I suggest to replace selectivity with sensitivity throughout the manuscript. The term sensitivity leaves the question of discrimination open.”

Thank you for the suggestion. We have replaced “selectivity” with “sensitivity” throughout the manuscript. For example the sentence that used to say: “*In order to localize the cortical regions uniquely selective to phonemic processing...*” now reads : “*In order to localize the cortical regions sensitive to phonemic processing* “. We deleted uniquely and replaced selective by sensitive.

“I wonder if the functional segregation between semantic and phonemic processing is really as categorical as figures 6A and B suggest. The choice of the colors may mislead the reader as diagonal is gray along all levels of sensitivity. Figure 6D suggests preference rather than categorical difference in most areas. It may be helpful to use a more continuous color scale (e.g. red, yellow, green) that lets the reader appreciate the degree of multiplexing. One advantage of the method used here is that it can quantify functional overlap.”

Thank you for the suggestion. We have changed the color scale to a continuous red-yellow-green.

“If no other restrictions prohibit it the authors should add a note on how the interested reader can obtain code and data to reproduce the results.”

Thank you for the reminder. Data availability and code availability sections are added.

Minor comments:

“p.19 “Our simulations (Methods), however, show that the capability of detecting fast neural activities underlying speech processing using fMRI is not necessarily constrained by this low temporal resolution as long as the fast speech events being studied are temporally sparse.” I do not think they need to be temporally sparse. They only need to be sufficiently variable at the right temporal scale. At least for low dimensional models. Sparseness may help to with the problem multicollinearity in big models.”

Yes, we agree with the reviewer that “sufficient variable at the right temporal scale” is the key. We have revised the contexts accordingly.

“p. 19 Was there any control to ensure the participants listened to the stories?”

We did not quantify the attentive state of the participants but they were all highly motivated subjects and during debriefing were all able to recall the stories.

“p.20 What is the "overall" template? Over all runs but within participant or over all subjects and runs?”

The overall template is obtained within participant/for each subject, over all runs. We have revised the text accordingly.

“p.21 what is the "overall" template? Over all runs but within participant or over all subjects and runs?”

The overall template is obtained within participant/for each subject, over all runs. We have revised the text accordingly.

“p.23 Typo R2measuring”

Thank you for pointing this out. We have revised it accordingly.

Reviewer #3

“The study is very well presented. Very well written and make use of high quality visuals (figures) to communicate their methods and results.”

Thank you.

“Despite the several positive remarks I have already mentioned, I find the study lacks novelty. It is true that it is interesting to study the continuum that speech comprehension encompasses (from acoustic perception to semantic access), and that brain differences

along this continuum may be related to how humans make sense of speech. However, other linguistic characteristics overlap with phonemic size, for example phonemic frequency, lexicality, semantic neighbors size, etc.”

We agree with the reviewer (and the field :-)) that other linguistic features clearly also play an important role both in phonemic segmentation. The novelty in our results might be more in degrees than in terms of a novel result - we find it quite intriguing to discover the relatively large region in the temporal cortex that shows significant representation of diphones relative to single phones or longer combinations. As we discussed these phonemic areas are certainly intertwined with lexical retrieval since short words can sometimes explain a large fraction of this sensitivity and semantic features can explain in the same regions' additional variance in the BOLD signal. Our approach and results open the door for future investigations on overlapping linguistic characteristics in order to refine this picture. As both you and reviewer 1 suggested, phonemic frequency might also be an interesting parameter to examine. We have added a section on its contribution in comparison to phonemic identity for the BOLD signal measured in these phonemic areas (see our response to reviewer #1)

“This study is very well thought, executed and written, but given the several other studies using the exact same stimuli and analysis, the current study would benefit from some novel aspect that could validate the author's conclusions.

For example: If subjects would also listen to a similar story in a language they don't understand, we should expect very different results in the access of semantic knowledge.”

Note that although we have used some of the same stimulus/responses that have been analyzed previously, this particular data set is larger (more subjects), including subjects with longer recording times. Also, in none of our previous analyses did we examine a combination of phonemic features as done here. This has allowed us to more systematically and directly assess the speech segmentation processes (albeit with limitations as mentioned in the previous remark). This is not something that has been done previously in an fMRI study as far as we know

However, we completely agree that experiments that compare and contrast responses to very different stimuli would be very useful. It is something that we are currently doing in a project that involves bilingual subjects. Using different languages will allow us to not only assess whether phonemic regions for different tongues converge but also whether phonemic processing also occurs without linguistic comprehension. The data published here for native english speakers listening to english stories makes such comparison possible.

We have added a comment along these lines in the discussion section of the revised manuscript.

“Lastly, VMs could also be used to explore the speech representations of bilingual or multilingual subjects listening to stories in multiple languages including some that subjects do not

understand. Such experiments would allow researchers to further distinguish phonemic segmentation and identification that occurs independently of understanding, or that might be common across multiple languages. These analyses would therefore be useful to also more clearly delineate the purely phonemic regions from those implicated in lexical retrieval, if such regions exist.”

“I also have a couple of remarks that could be better explained in the manuscript:

1- The cross-validation method involves cutting epochs of 20 seconds (10 TRs) and permuting them to assess chance level in fMRI signal prediction based on the training model parameters. In Heer 2017, 40 seconds were used for the same goal. WHY using a smaller window size for the permutations? And why now permute the labels instead (i.e., the story ID). I fear that this strategy to permute windows of data may have consequences to the low frequencies that characterize the BOLD signal (as the authors elude in the method's section), by introducing abrupt signal changes that the original (un-permuted) signal doesn't have. “

Thank you for your comments. These are valid concerns that we also appreciate as the reviewer knows but that were not well addressed in this manuscript. In the revised version, we permuted the feature labels and reran all the analysis. Please refer to the response toward the second comment of the reviewer 2.

“2- WHY exactly is the R2 used for model validation instead of the r? If I understand correctly, the model is employed to predict the fMRI response, and correlations verify whether the prediction and real signal are similar. If so, one expects positive correlations (r), and R2 hides this. I understand that R2 speaks for variance explained and it should still be expected that the permuted data should have less variance explained. So, my remark (question) is more on the conceptual aspect of how to employ this encoding models. Positive correlations are meaningful and negative correlations should not be meaningful. If this is correct, by hiding the sign of the correlation (say in the brain maps) may create clusters of voxels with opposite r signs.”

There are a couple of good reasons for choosing the cross-validated (CV) R^2 instead of the Pearson correlation coefficient. First, R^2 is directly proportional to the metric used to fit the weights in the linear encoding model, where the error function is based on the sum of squared errors (e.g. Du pre, 2022). The weights that yield the minimum R^2 are unique while there are many that can give similar Pearson r values (for example all the ones where the regression weights are scaled by the same factor). Second, the raw CV R^2 (before normalization by the explainable variance or ceiling value) can capture both the quality of the prediction and the noise level which can be a good metric. The absolute scale of the prediction (lost in the Pearson r) is particularly informative in ridge regression. In voxels with high predictive power (as detected during cross-validation), the ridge regularization is low, and the prediction scale is high. But in voxels with no predictive power (as detected during cross-validation), the ridge

regularization is high, and the prediction scale is low (close to zero). This prediction scale can be lost in the r-values.

To more specifically address the reviewer's concern on losing the "sign" of the prediction. The analytical solution to the ridge regression is based on the signed cross-correlation term between the stimulus and the response - the same sign that is used in the Pearson correlation coefficient. This is also a direct consequence of minimizing the sum of errors. Thus, as long as the cross validated (and unbiased estimate) of R^2 is significant, the corresponding Pearson correlation coefficient must be positive.

Note that the unbiased estimate of the CV R^2 can also be positive or negative. It can have large negative values, even when there is predictive power, if the model was overfit. When there is no predictive power in the features, the expected value of the CV R^2 is zero with a symmetric distribution that is centered around that zero point.

To illustrate the relationship between significant R^2 and Pearson's r, we show here the scatter plot of the square root of R^2 and Pearson's r for all significant voxels assessed by our permutation test. It reveals that the coefficient of determination R^2 and the Pearson's r provide a very similar quantification of the goodness of fit of the model and that R^2 is a slightly more conservative estimate of that predictive power..

Reviewer #1 (Remarks to the Author):

The authors have addressed my previous concerns very well. They have added several new analyses to strengthen their conclusions. In my opinion, the paper is suitable to be published in Nature Communications.

Reviewer #2 (Remarks to the Author):

The authors have sufficiently addressed my points. I recommend the manuscript for publication.

Reviewer #3 (Remarks to the Author):

The authors have fully answered my remarks. Additionally, the authors made changes in their analysis pipeline regarding the permutation method, which constituted my main methodological concern. I agree with the changes in the manuscript.